# Evaluating the hydrological consistency of evaporation products using satellite-based gravity and rainfall data

Oliver López[1], Rasmus Houborg[1], Matthew F McCabe[1]

[1]Water Desalination and Reuse Center, Division of Biological and Environmental Sciences and Engineering, King Abdullah University of Science and Technology, Thuwal, 23955-6900, Saudi Arabia

*Correspondence to*: Oliver López (oliver.lopez@kaust.edu.sa)

**Abstract.** Advances in space-based observations have provided the capacity to develop regional- to global-scale estimates of evaporation, offering insights into this key component of the hydrological cycle. However, the evaluation of large-scale evaporation retrievals is not a straightforward task. While a number of studies have intercompared a range of these evaporation products by examining the variance amongst them, or by comparison of pixel-scale retrievals against ground-based observations, there is a need to explore more appropriate techniques to comprehensively evaluate remote sensing based estimates. One possible approach is to establish the level of product agreement between related hydrological components: for instance, how well do evaporation patterns and response match with precipitation or water storage changes. To assess the suitability of this "consistency" based approach for evaluating evaporation products, we focused our investigation on four globally distributed basins in arid- and semi-arid environments, comprising the Colorado River basin, Niger River basin, Aral Sea basin and the Lake Eyre basin. In an effort to assess retrieval quality, three satellite-based global evaporation products based on different methodologies and input data, including CSIRO-PML, MOD16 and GLEAM, were evaluated against rainfall data from GPCP along with GRACE water storage anomalies. To ensure a fair comparison, we evaluated consistency using a degree correlation approach after transforming both evaporation and precipitation data into spherical harmonics. Overall we found no persistent hydrological consistency in these dryland environments. Indeed, the degree correlation showed oscillating values between periods of low and high water storage changes, with a phase difference of about 2-3 months. Interestingly, after imposing a simple lag in GRACE data to account for delayed surface runoff or baseflow components, an improved match in terms of degree correlation was observed in the Niger River basin. Significant improvements to the degree correlations (from ~0 to about 0.6) were also found in the Colorado River basin for both the CSIRO-PML and GLEAM products, while MOD16 showed only half of that improvement. In other basins, the variability in the temporal pattern of degree correlations remained considerable and hindered any clear differentiation between the evaporation products. Even so, it was found that a constant lag of two months provided a better fit compared to other alternatives, including a zero lag. From a product assessment perspective, no significant or persistent advantage could be discriminated across any of the three evaporation products in terms of a sustained hydrological consistency with precipitation and water storage anomaly data. As a result, our analysis has implications in terms of the confidence that can be placed in

independent retrievals of the hydrological cycle, raises questions on inter-product quality, and highlights the need for additional techniques to evaluate large-scale products.

## 1 Introduction

Space-based observations of the Earth system have provided the capacity to retrieve information across a wide range of land surface hydrological components and an opportunity to characterize terrestrial processes in space and time. Indeed, remote sensing offers a number of independent means with which to retrieve various components of the hydrological cycle (e.g. rainfall, soil moisture, evaporation, terrestrial storage). Progress in satellite-based observation of the Earth system has enabled the characterization of land surface hydrological components and an improved representation of terrestrial processes

(Famiglietti et al., 2015). Dedicated space missions such as the Gravity Recovery and Climate Experiment (GRACE) (Tapley et al., 2004b), the Global Precipitation Measurement Mission (GPM) (Hou et al., 2014) and a suite of microwave-based soil moisture platforms (Liu et al., 2012), represent important efforts that have contributed to these advances. Considering the spatial advantage that space-based observations have over ground-based measurements, there has been a proliferation of regional to global-scale data products, providing knowledge on the multi-scale behaviour and patterns of

hydrological states and fluxes useful for enhanced process description (Stisen et al. 2011). However, one of the challenges of space-based remote sensing is how to characterize the degree to which these products represent realistic estimates of the underlying variables they attempt to retrieve.

Terrestrial evaporation (E), comprising the sources of soil and canopy evaporation together with plant transpiration, plays a

key role in the water cycle as a linking mechanism between the surface and the atmosphere (Mueller et al., 2011). Unlike microwave or radiative emissions from the surface or atmosphere, which can be used to inform upon soil moisture, surface temperature or rainfall, evaporative fluxes provide no directly observable trace that can be detected from satellites and are instead estimated through interpretive or empirical models (Jimenez et al., 2011; Ershadi et al., 2014). Recently, several of these models have been used to develop global-scale evaporation products by combining satellite observations of surface

variables with meteorological and other ancillary data (McCabe et al., 2016; Miralles et al., 2016). When ground-based flux observations are available, they can be used for calibration and evaluation (McCabe et al. 2005a): but large-scale assessment is inevitably constrained by the lack of distributed and representative in situ networks to comprehensively assess simulations as well as the inherent uncertainty associated with these observations (Jana et al., 2016). Some recent evaluation efforts have sought to estimate the uncertainty of satellite-based evaporation products, as well as those from land surface model and

reanalysis data, in terms of the variance amongst the products (Mueller et al., 2011; Jimenez et al., 2011; Long et al., 2014). These and related attempts have shown that no single evaporation model consistently outperforms any other, whether applied at local (Ershadi et al., 2014) or global scales (Miralles et al., 2016), even in cases where the same input data has been used.

Considering the issue of spatial mismatch and model variability, it seems inappropriate to assess these large-scale products via direct comparison to in situ data alone. Moreover, the quality of any satellite-based product should not be judged solely on its agreement with potentially unrepresentative point-scale approaches. Central to this challenge is the issue of scale, a consequence of both a lack of abundant high-quality in situ data and the fact that there is an inevitable scale mismatch between ground- and satellite-based observations (McCabe et al. 2005b). To compensate for this, it is important that a range of methods be used to evaluate the large-scale implementation of evaporation models.

Beyond the assessment of evaporation models, a limited number of studies have sought to quantify large-scale water budgets using either satellite observations alone (Sheffield et al., 2009) or through a combination of satellite observations and data assimilation (Pan and Wood, 2006; Pan et al., 2008; Sahoo et al., 2011; Pan et al, 2012). While some of these studies (Sheffield et al., 2009; Gao et al., 2010) evaluate water budget closure by comparing the residual of the water budget (i.e. inferred runoff) with measured runoff, others aim to provide merged or observation constrained estimates of the water cycle components, with estimates of uncertainty given in terms of the variability among the products (e.g. Long et al., 2014). The results of these studies have generally illustrated large water budget closure errors, focusing on the temporal scale and invoking the use of a hydrological model to guide analysis or force closure, rather than being solely observation driven assessments. Observation-only studies are important, as they provide an unbiased perspective not just on hydrological closure, but also allow for a first-order examination of the underlying agreement between component variables. However, rather than just comparing the uncertainties between evaporation products and other hydrological components (which are poorly defined), there is still a need for alternative assessment techniques that exploit the inherent connection between hydrological variables at both temporal and spatial scales. One approach to determine this is to evaluate the hydrological consistency between observed products (McCabe et al., 2008). The term hydrological consistency refers to the spatial and temporal match that should exist between independent observations of hydrological states and fluxes, based upon physical considerations. It is a concept that encompasses the expectation of water cycle behavior and mass balance: that is, changes in one term should be reflected in related variables, both spatially and temporally. For instance, a rainfall event should result in an observable change in soil water storage and a consequent increase in evaporative flux, which in turn should reduce the available soil moisture. This relatively simple concept has been explored in the recent past, including in efforts to improve precipitation events by employing cloud detection methodologies (Milewski et al., 2009); using soil moisture changes to infer precipitation amounts (Brocca et al., 2014); examining the connection between soil moisture state and changes in atmospheric variables such as humidity and sensible heat flux (McCabe et al., 2008); as well as in assessments of land−atmosphere coupling between observations and reanalysis data (Ferguson and Wood, 2010).

In considering these earlier contributions, there remains a need to determine whether the basic idea of hydrological consistency can be realistically extended to explore the agreement between independent global-scale satellite-based hydrological products. To examine this question, it makes sense to focus on catchments that have relatively simple

hydrological interactions, as they represent natural laboratories within which the evaluation of large-scale products and the concept of hydrological consistency can be reasonably undertaken. For example, Wang et al. (2014) evaluated the level of agreement between three satellite-based hydrological cycle variables over arid regions in Australia, where surface and subsurface runoff were minimal. With a sufficiently low runoff component, a lack of snow accumulation, and a relatively

strong coupling of precipitation and evaporation components, arid and semi-arid environments represent potential candidates within which to undertake such process assessments. Recognizing the need to advance a more comprehensive evaluation strategy for remote sensing retrievals, this study seeks to explore the hydrological consistency within a number of basins where hydrological processes are relatively simple, i.e. reflecting the conditions described above. Our analysis constitutes a framework for assessing the utility of hydrological consistency to evaluate remotely sensed hydrological products. We

undertake the analysis over four large river basins within arid and semi-arid environments distributed across the globe, with study regions comprising the Colorado River basin in North America, the Niger River basin in Africa, the Aral Sea basin in Asia, and the Lake Eyre basin in Australia.

In compiling datasets with which to evaluate and differentiate between candidate evaporation products, a number of product

specific considerations needed to be accounted for. Total water storage estimates, which comprise the summation of groundwater, soil moisture, snow, surface water, ice and biomass, were derived from anomalies in the gravity field from GRACE satellites (Tapley et al., 2004b). As any continuous function on a sphere, the gravity field can be represented as an expansion in spherical harmonics, which form a complete set of basis functions in the sphere: similar to the way in which a Fourier series expansion uses sines and cosines as basis functions. Unlike precipitation and evaporation products (and most

other hydrological remote sensing variables), it is problematic to directly compare spatial maps of GRACE water storage data with other spatially distributed hydrological variables (Tapley et al., 2004a), since GRACE data are usually filtered in the spectral domain. While scaling the GRACE data to account for differences due to filtering has been proposed as a solution to this problem (Landerer and Swenson, 2012), it has recently been shown to affect results in certain cases (Long et al., 2015), including over arid regions. Given this restriction, we implement an alternative approach in which the

precipitation and evaporation fields are transformed into spherical harmonics in order to remove the impact (and model dependence) of this scaling term. Such an approach allows for a more reasonable and equivalent intercomparison of hydrological variables, and represents a novel aspect of this work. Further details describing this process are presented in Section 3.

The overall objective of this study is to evaluate the hydrological consistency of three global scale satellite-based evaporation products against remotely sensed retrievals of precipitation and terrestrial water storage across a selection of basins that exhibit relatively well-defined hydrological interactions. Throughout this analysis we aim to determine whether the hydrological consistency concept can expand the range of evaluation metrics used to assess large-scale hydrological data sets such as evaporation, and enable some differentiation of relative product quality to be made.

## 2 Data sources and study regions

A range of globally distributed large-scale data sets derived primarily from satellite observations were used in this analysis. The study period, encompassing the years between 2003-2011, was based upon the availability of GRACE data and several recently developed global-scale evaporation products. In the following paragraphs we briefly describe the sources and nature of the data used in this contribution.

### 2.1 GRACE water storage anomalies

GRACE water storage estimates have been used in a myriad of studies exploring the indirect groundwater response across many different spatial and temporal scales (Swenson et al., 2008; Rodell et al., 2009; Famiglietti et al., 2011; Sun, 2013; Voss et al., 2013). The accuracy of GRACE terrestrial water storage anomalies (TWSA) is related to the number of degrees to which the gravity field is solved for in spherical harmonics (Swenson and Wahr, 2002) and an approximate global averaged accuracy of 20 mm.month$^{-1}$ has previously been proposed (Wahr and Velicogna, 2006). Water storage anomalies (2003-2011) were computed using GRACE (release 05) monthly spherical harmonic coefficients representing the gravity field, processed at the University of Texas Center for Space Research (UTCSR). The gravity field is usually described in terms of the geoid: an equipotential surface that is defined to correspond to the mean sea level over the oceans (Swenson and Wahr, 2002). The geoid is usually approximated as a linear combination of spherical harmonics, given that these represent solutions to the Laplace equation that describes the relation between the gravitational potential and the geoid. The approximation is of the form:

$$f(\theta, \phi) \approx \sum_{l=0}^{l_{max}} \sum_{m=0}^{l} \widetilde{P_{lm}}(\cos(\theta))(C_{lm} \cos(m\phi) + S_{lm} \sin(m\phi)), \tag{1}$$

where $\tilde{P}_{lm}(\cos\theta)$ are the normalized associated Legendre functions, $\theta$ corresponds to the colatitude (the complementary angle to the latitude), $\phi$ to longitude, $C_{lm}$ and $S_{lm}$ are the spherical harmonic coefficients of degree l and order m, and $l_{max}$ is the truncation degree. The total number of coefficients is given by $((l_{max} +1)^2 + l_{max}+1)/2$, while the resolution (the scale of the smallest feature of the gravity field that can be resolved using $l_{max}$ coefficients) is approximately $\pi a/l_{max}$ (where a is the Earth's radius). The data product used in this study (processed by UTCSR) contains coefficients up to $l_{max} = 60$, i.e. a total number of 1891 coefficients, with an approximate resolution of 333 km. The full description of the process to transform the gravity field anomalies into water storage anomalies is described in Wahr (1998) and Swenson and Wahr (2002).

GRACE data contains two types of errors (correlated and random) that need to be filtered before translating the data into water storage anomalies. Correlated errors are known to contaminate the signal in the form of north-south oriented stripes. A "de-striping" filter was applied to the coefficients (Swenson and Wahr, 2006; Duan et al, 2009) in order to remove this source of error. An isotropic filter (Gaussian filter with radius of 300 km) was then used to remove random errors (Swenson and Wahr, 2002). Furthermore, it is a usual practice to replace the degree 2 coefficients with a more reliable estimate from a

low-degree model of the gravity field calculated using satellite laser ranging (Cheng et al., 2011;Cheng et al., 2013). While the effect that the filters have on the true geophysical signal is not known a priori, an indirect measure can be obtained by applying the filter to a synthetic water storage variation from a land surface model (LSM). This method has been used to obtain scaling factors for GRACE data in order to restore the signal (it has been observed that the filters typically reduce the signal) before using the GRACE data with other hydrological variables (Landerer and Swenson, 2012). Long et al. (2015) evaluated the impact of different land surface models on the scaling factor and showed that the impact was greatest in arid regions. To avoid this potential element of uncertainty in our study, which is focused on arid regions, we instead transformed the other water cycle components (i.e. evaporation and precipitation) into spherical harmonics, using an approximation similar to equation 1. The effect of the filters is therefore incorporated directly into the other hydrological components in spherical harmonics.

## 2.2 Evaporation products

Several satellite-based evaporation products have been developed over the last decade, based on a range of modeling schemes (Mu et al., 2011; Leuning et al. 2008; Miralles et al., 2011a) and global-scale input data. Given the importance of evaporation within studies of the global energy and water cycle, considerable effort has been directed towards accurately reproducing its spatial and temporal variability, with comprehensive reviews of various approaches to do this provided by Kalma et al. (2008) and Wang and Dickinson (2012). Here we employ a range of global evaporation datasets, which are briefly described in the following paragraphs and summarized in Table 1.. To ensure consistency with the GRACE data, the evaporation products were aggregated from daily (or 8-daily in the case of MOD16) to monthly estimates, centered on the dates specified in the GRACE monthly gravity field solutions. In the aggregation from daily to monthly data, pixels that presented missing data for more than 20% in a given month were not included in the calculation.

### 2.2.1 MOD16

Cleugh et al. (2007) developed an algorithm for large-scale evaporation monitoring based on the Penman−Monteith (PM) equation, using meteorological forcing data and a surface resistance linearly modeled through remotely sensed leaf area index (LAI), as measured by the MODerate resolution Imaging Spectroradiometer (MODIS). Improvements to this approach (Mu et al., 2007; Mu et al., 2011) led to the development of the MODIS Global Evapotranspiration product (MOD16), a three-source scheme used for terrestrial land flux estimation. In MOD16, the linearization of the surface resistance is specified for each biome separately via a look-up table, with the evaporation calculated for daytime and nighttime conditions. Other adjustments incorporated into MOD16 include soil heat flux calculation, distinction of dry and wet canopy, as well as moist and wet soil, and improvements to the aerodynamic resistance. The MOD16 product comprises transpiration, evaporation from the soil and wet canopy, as well as total evaporation calculated as the sum of these three components. Each component is weighted-based on the fractional vegetation cover, relative surface wetness and available energy. Inputs to the model include net radiation ($R_n$), air temperature and humidity, as well as LAI and vegetation

phenology. Importantly, it does not require wind speed, precipitation or soil moisture data, making it a relatively parsimonious model in terms of input requirements. In this study, we used the actual evaporation (AET) product from MOD16 (Mu et al., 2011) with 8-day temporal resolution and 1 km resolution in the sinusoidal projection. The product was reprojected onto a 0.05° regular grid using the MODIS Reprojection Tool (MRT) before transformation into spherical 5  harmonics, as described in Section 3.1. Further details on the modeling basis behind the MOD16 product can be found in Mu et al. (2013), Ershadi et al. (2014) and Michel et al. (2016).

### 2.2.2 CSIRO-PML

In parallel to the PM-Mu model, Leuning et al. (2008) introduced improvements to the Cleugh et al. (2007) algorithm, 10  resulting in the two-source Penman−Monteith−Leuning (PML) model. An important new feature of the PML approach was a biophysical algorithm for the calculation of the surface resistance, which was previously calculated as LAI multiplied by a constant $c_L$ (Cleugh et al., 2007). The new parameterization of the surface resistance in the PML model was optimized using data from 15 globally distributed flux station sites, with two key parameters identified: the maximum stomatal conductance ($g_{sx}$) and the ratio of actual to potential evaporation at the soil surface. Zhang et al. (2010) developed a method to further 15  optimize the spatial variability of these two parameters (i.e. at each grid pixel) using gridded meteorological data and a simple Budyko-curve hydrometeorological model developed by Fu (1981) that includes precipitation and available energy as inputs. Mean annual evaporation for each grid pixel is calculated using the Fu model and gridded meteorological data. The value of $g_{sx}$ is optimized using a non-linear least square regression based on the difference between the PML and the Fu model. Interestingly, the Fu model is calibrated by comparing the output evaporation with the residual of precipitation and 20  runoff i.e. by assuming negligible annual water storage changes and groundwater inflow and outflow. Zhang et al. (2012) used this approach to develop a global gridded terrestrial evaporation product (hereafter referred to as CSIRO-PML; Zhang, 2014, personal communication) with a 0.25° resolution (in this study, we used the actual evaporation product). They used gridded meteorological data from diverse sources, including vapor pressure and temperature from the Climate Research Unit (New et al., 2000), LAI and land cover type from Boston University (Ganguly et al., 2008), precipitation from the Global 25  Precipitation Climatology Centre (GPCC, version 4; Schneider et al., 2011), and radiation data from the Global Energy and Water Cycle Exchanges (GEWEX) Surface Radiation Budget (Gupta et al., 2006).

### 2.2.3 GLEAM (v2A)

The Global Land Evaporation: the Amsterdam Methodology (GLEAM) (Miralles et al., 2011a) is a satellite-based model 30  developed to estimate evaporation at a global scale. In this approach, rainfall interception loss is evaluated using an analytical model (Gash, 1979) as a first step. GLEAM then employs the Priestley−Taylor equation to calculate the potential evaporation of bare soil and vegetation components (both short and tall canopy), with values constrained to actual

evaporation via application of a stress factor. The stress factor is calculated using vegetation optical depth from a combination of different satellite passive microwave observations using the Land Parameter Retrieval Model (Liu et al. 2013). GLEAM also has the capacity to explicitly calculate sublimation of snow covered surfaces (Takala et al., 2011) as well as open water evaporation. Satellite observations of surface soil moisture can be assimilated using a Kalman filter assimilation approach to estimate the moisture profile over several soil layers. Here we employ version 2A of GLEAM (Miralles, 2014, personal communication), which uses a combination of satellite, ground and reanalysis input data. Precipitation is obtained from the Climate Prediction Center Unified data set, consisting of data from over 30,000 stations (CPC-Unified, Joyce et al., 2004). The radiation product used in this version of GLEAM is the European Center for Medium-Range Weather Forecasts (ECMWF) ERA-Interim meteorological reanalysis product (Dee et al., 2011). In this version of GLEAM, surface soil moisture data from the Water Cycle observation Multi-mission Strategy Climate Change Initiative (WACMOS-CCI) merged product (from a combination of several passive and active microwave products) is assimilated (Liu et al., 2012), while air temperature is derived from both the International Satellite Cloud Climatology Project (ISCCP) and the Atmospheric Infrared Sounder (AIRS) (Rossow and Dueñas, 2011). Further details of the model can be found in Miralles et al. (2010), Miralles et al. (2011a) and Miralles et al. (2011b).

## 2.3 Precipitation data

Global daily precipitation (P) estimates derived from multi-satellite observations for the period 2003-2011 were obtained from the Global Precipitation Climatology Project (GPCP) (Huffman et al., 2001), the official World Climate Research Program (WCRP) Global Energy and Water Cycle Exchanges (GEWEX) product. The GPCP product is a merged precipitation analysis combining information from microwave, infrared, and sounder data observed by a constellation of international precipitation-related satellites (Huffman 1997; Huffman 2001; Adler 2003). The estimates from microwave and infrared data are based on the Threshold-Matched Precipitation Index (TMPI). The combined satellite-based product is corrected by rain gauge analysis where data is available (i.e. the intermediate product from the Global Precipitation Climatology Center; Schneider et al., 2011). Over many areas of the world, the GPCP product represents one of the best available sources of precipitation data and has been previously used in soil moisture- and evaporation-based analyses (Crow, 2007; Miralles et al., 2011a). In this study, we used the daily product (GPCP 1DD) and converted daily values to monthly estimates, centered on the dates provided in GRACE monthly gravity field solutions. Pixels were assigned as missing data when more than 20% of the month was missing (on a per-pixel basis).

## 2.4 Runoff data

While runoff data was not used explicitly in the consistency analysis presented in this manuscript, simulated runoff data was compared to precipitation and evaporation observations in order to evaluate the assumption of a relatively simple hydrological system in the study basins. Surface runoff, sub-surface runoff and snowmelt were derived from the NOAH land

surface model included in the Global Land Data Assimilation System (GLDAS) (Rodell et al., 2004). GLDAS uses global satellite and ground-based observational products to obtain optimal estimates of land surface states and fluxes from land surface models using data assimilation techniques. Although these values were not constrained with ground estimates and thus may contain biases, as noted, runoff values were only used to provide an assessment of runoff against the observed

precipitation and evaporation data. The version of the product used in this study (GLDAS-2.0) is forced with meteorological data from the Princeton University forcing data set (Sheffield et al., 2006) and is available at 1° resolution from 1948 to 2010.

### 2.5 Selection of study basins

The study basins were targeted primarily on their climate classification, with river basins in regions with a predominantly

arid or semi-arid climate preferentially selected. This criterion was established in order to seek a relatively simple hydrological system (i.e. constrain the range of possible hydrological interactions), thereby maximizing the conditions under which hydrological consistency between evaporation and precipitation and water storage changes might be achieved. A Köppen classification map, generated using data sets from the Climatic Research Unit and the Global Precipitation Climatology Centre up to 2006 (Kottek et al., 2006), was used to identify arid and semi-arid regions. The basins were

selected from a set of 405 globally distributed river basins provided by the Global Runoff Data Centre (GRDC) and derived from flow direction data of the HYDRO1k Elevation Derivative Database, developed at the U.S. Geological Survey (USGS). A threshold of 50% areal extent containing any of the arid Köppen climates (BWk, BWh, BSk or BSh) was used to select potential basins. Secondary criteria for basin selection from the GRDC data set focused on size, geographical distribution and amplitude and trends in the water storage variations. In terms of the size of the basin, a smaller size would more likely

satisfy the assumption of a relatively simple hydrological system. However, due to the coarse resolution of GRACE data (see section 2.1), this requirement had to be compromised. Given these considerations, four basins were selected as focus regions of study: the Colorado River basin (CRB) in North America, the Niger River basin (NRB) in Africa, the Aral Sea basin (ASB) in Asia and the Lake Eyre basin (LEB) in Australia (Figure 1).

Figure 2 shows the spatially averaged hydrological fluxes over the study basins, including the sum of surface, subsurface and snowmelt runoff (Q) derived from the GLDAS NOAH version 2 monthly product (Rodell et al., 2004). Q is included in these figures to establish the extent to which a major assumption of the study (i.e. a simple water budget) is met across each of the study regions. Although predominantly arid, with a combination of hot arid desert and cold arid steppe climate classifications, both the Colorado River basin and the Aral Sea basin contain a snow component. Snowmelt in these two

regions plays an important role in the water cycle, particularly in the delivery and redistribution of water to other areas of the basin. Therefore, hydrological consistency might not be satisfied completely in these regions using our simple water budget assumption for some periods. Likewise, the Niger River basin also has a runoff component that is close in magnitude to evaporation, but we assume that it will not affect the spatial distribution of water storage anomalies. In the Lake Eyre basin

we expect that the limited and sporadic runoff component will not have a significant effect on the hydrological consistency analysis undertaken there.

Even though these four basins were preselected based upon their location within dryland systems (Wang et al., 2012), they reflect a range of trends in water storage and precipitation. For example, the Colorado River basin experienced intervals of wet and dry periods, while the Niger River basin exhibits a small but steady increase in water storage with a clear seasonal variability in both water storage and precipitation. Meanwhile, the Aral Sea basin experienced a significant loss of water during the study period (-8.2 mm.yr$^{-1}$), in line with the historical depletion of this inland sea in response to increased agricultural productivity (Zmijewski and Becker, 2014). The Lake Eyre basin showed a marked increase in precipitation during the end of the study period (2009-2011), with a corresponding increase in water storage during the following years, reflecting the larger scale hydrometeorological conditions affecting that region (Boening et al., 2012).

## 3 Methodology

In order to provide a meaningful spatial evaluation of the hydrological consistency between the data sets (i.e. at sub-basin scale) and to ensure that a fair comparison between GRACE data and satellite products could be undertaken, the analysis was carried out in spherical harmonics. The effects of the de-striping filter (see Section 2.1) are incorporated into the analysis directly instead of relying on a land surface model, the choice of which can severely impact the results of our analysis in arid regions (Long et al., 2015). In this section, we present a detailed account of how the transformation was carried out, as well as how the actual evaluation of hydrological consistency is performed in spherical harmonics.

### 3.1 Spherical harmonic analysis of evaporation and precipitation data sets

The spherical harmonic analysis refers to the process of solving equation 1 for a set of coefficients $C_{lm}$ and $S_{lm}$ up to an approximation of degree $l_{max}$. Several computational packages are available to perform this type of analysis. Here we used a FORTRAN program developed by Wang et al. (2006), which is suited for regularly gridded regional and/or global non-smooth data sets. The program can also perform spherical harmonic synthesis, which is the inverse transformation (i.e. from coefficients to spatial data). Figure 3 presents an example of the transformation based on the gridded CSIRO-PML data for April 2003. Because all data sets are evaluated up to the same degree $l_{max}$, any differences due to the mismatch in the resolution of the products are eliminated after spherical harmonic analysis and synthesis. After this process, we generated three P−E anomaly data sets, i.e. one for each E product. Next, we applied the de-striping and Gaussian filters to account for the effect that these have in GRACE TWSA data (see section 2.1).

## 3.2 Regional spherical harmonic analysis

In the analysis so far, the computed spherical harmonic coefficients are undertaken at the global scale (e.g. Figure 3). In order to evaluate the hydrological consistency of the study regions (Figure 1), the data needs to be masked for the particular study basins. In Swenson and Wahr (2002), an exact averaging kernel is defined as a function with a value of 1 inside the boundaries of a region and 0 outside. To isolate the GRACE signal, an approximated averaging kernel was computed in spherical harmonics and convolved with the Gaussian filter in order to obtain a spatially averaged value of the TWSA (at the basin scale). In this study, we instead compute the spherical harmonic from the product of the global data sets (e.g. TWSA or P-ET) with the averaging kernel following Eq. (2):

$$f^b(\theta, \phi) = f^g(\theta, \phi)\vartheta(\theta, \phi), \tag{2}$$

where $f^b(\theta,\phi)$ is the isolated regional data, $f^g(\theta,\phi)$ is the global data set and $\vartheta(\theta,\phi)$ is the averaging function. The relation in spherical harmonics is given by Eqs. (3-4):

$$f^b = \Sigma_{j_1,m_1} \sum_{j_2,m_2} f^g_{j_1,m_1} \vartheta_{j_2,m_2} Q^{jm}_{j_1m_1j_2m_2}, \tag{3}$$

$$Q^{jm}_{j_1m_1j_2m_2} = \sqrt{\frac{(2j_1+1)(2j_2+1)}{4\pi(2j+1)}} C^{j0}_{j_10j_20} C^{jm}_{j_1m_1j_2m_2}, \tag{4}$$

where $C^{jm}_{j_1m_1j_2m_2}$ are the Clebsch−Gordan coefficients (Martinec, 1989). We used the program developed by Martinec (1989) to mask the three global P-ET data sets (as well as GRACE data) over the four study regions (Figure 1).

## 3.3 Evaluating spatial agreement in the spherical harmonics of two data sets

The spatial agreement between two data sets can be evaluated using spherical harmonic coefficients by computing the degree correlation measure (Arkani-Hamed, 1998; Tapley et al., 2004a) following Eq. (5):

$$r_l = \frac{1}{\sigma^{(A)}_l \sigma^{(B)}_l} \sum_{m=0}^{l} (C^{(A)}_{lm} C^{(B)}_{lm} + S^{(A)}_{lm} S^{(B)}_{lm}), \tag{5}$$

where $\sigma^2_l$ is the degree variance given in Eq. (6):

$$\sigma^2_l = \sum_{m=0}^{l} (C^2_{lm} + S^2_{lm}), \tag{6}$$

The degree correlation measure is computed for every degree (l), and therefore we can in principle evaluate the hydrological consistency at different length scales (i.e. sub-basin variability). As noted earlier, GRACE data is limited in resolution by $l_{max}$ =60, or to approximately 330 km. In practice however, the de-striping filter removes all coefficients larger than 40 and as such we are limited to length scales of about 500 km and larger. The smallest basin in this study is the Colorado River, covering an area of about 640,000 km$^2$. Based on this area, we can set a limit for the approximate largest spatial scale relevant to our study as 800 km, corresponding approximately to degree 25.

**4 Results**

**4.1 Assessing the consistency of evaporation products**

An examination of the evaporation data sets indicates that there are evident differences across the various products in each of the studied basins (see Figure 2). In general, MOD16 simulates lower flux estimates when compared against both CSIRO-
PML and GLEAM, a feature that has been noted in a number of recent global intercomparison studies (McCabe et al., 2016; Michel et al., 2016; Miralles et al. 2016). There are also clear differences in terms of the variability in the temporal response of the models, although CSIRO-PML and GLEAM show a greater level of agreement in terms of amplitude and timing, if not in absolute values. For example, during the wet period of 2004-2005 in the Colorado River basin, the response to precipitation reflected in MOD16 was far more rapid than either CSIRO-PML or GLEAM displayed. Of some concern is
that CSIRO-PML is larger than precipitation during much of the study period in both the Colorado River and the Lake Eyre basins, immediately negating any type of hydrological consistency analysis. In the Niger River basin, there is more consistent agreement between the evaporation products, indicating greater confidence in the retrievals of evaporation in this region. For the Aral Sea basin, the discrepancies in E are similar to those obtained for the Colorado River basin, with an obvious phase shift in CSIRO-PML and GLEAM observed relative to MOD16. This may reflect complexities in evaporation
modeling due to the intermixed climate zones in the region caused by differences in land surface parameters. In the Lake Eyre basin, there are differences in amplitude but not in the temporal behavior of E.

Overall, even from a qualitative perspective, there are clear challenges in developing a hydrological consistency approach over these comparatively "simple" basins. Indeed, this has been demonstrated in other studies using either satellite data
alone, or a combination of satellite and ground data. While it is not the intent of the current work to explore the error characterization of these different evaporation models based on hydrological closure, the techniques being used to evaluate product consistency should provide some insight into retrieval quality: at least relative to the other hydrological products (precipitation and gravity-based water storage changes) that the evaporation is being compared against. These ideas are explored more quantitatively in the following sections.

**4.2 Basin-scale assessment**

In this section, we examine the spatial and temporal patterns of the degree correlations between water storage variations (TWSA) and P-E anomalies. Figure 4 to Figure 7 present the results of this assessment across each of the four large-scale basins. For each of the figures, time series of the spatial average TWSA and P-E anomalies are shown in order to compare their trends with the temporal behavior of the degree correlation ($r_l$). In these figures, the degree correlation is a measure of
the spatial agreement between the two fields being compared (i.e. TWSA and P-E anomalies), assuming that other outflow components such as surface runoff (assumed to be minimal in these basins) do not directly affect these spatial patterns. This comparison is helpful in determining whether the cause of trends in water storage variations (either natural or anthropogenic)

influence the analysis of hydrological consistency e.g. do the degree correlations behave differently during wet or dry periods, or when storage changes are driven by natural or anthropogenic causes? In these figures, the response of the degree correlations is shown in time across the x-axis and in the spectral domain along the y-axis, for each of the three evaporation products.

### 4.2.1 Colorado River basin

The start of the study period (2003) coincided with the end of an intense multi-year drought in the Colorado River basin (Scanlon et al., 2015). During the wet period of 2004-2005 (Figures 2), the basin showed a corresponding increase in TWSA (see Figure 4), although with a delay in time of two to three months. During this time of increase in TWSA, there was a corresponding increase in $r_l$ (up to 0.9 for l=25 and 0.8 for l=40) until TWSA reached its peak value (November 2004 – February 2005), after which $r_l$ decreased and showed negative values (similar, but negative, i.e. -0.9 for l=25; -0.8 for l=40) during the TWSA decrease. During the dry period (i.e. 2008-2009), TWSA is correspondingly lower, but oscillating out of phase with P-E anomalies (about 2 months of lag). There seems to be a stronger relationship between the oscillation of TWSA and degree correlations during the wet and dry periods than for other catchment conditions, where the variations of $r_l$ appear random. In general, the degree correlations for small degrees have larger amplitudes than those for large degrees. This spatial disagreement in correlation might be related to the spatial and temporal distribution of runoff in the basin, since a large portion of the runoff comes from snowmelt originating in the upper portions of the basin (Scanlon et al., 2015). Differences in absolute values and in the temporal distribution of E (especially with the MOD16 product) were evident in the degree correlation images in Figure 4. However, they did not have a significant impact on the analysis in the sense of demonstrating any product advantage or disadvantage relative to the other evaporation products, at least in terms of their hydrological consistency.

### 4.2.2 Niger River basin

The TWSA in this Niger River basin was characterized by an overall steady increase (5.79 mm.yr$^{-1}$) with clear seasonal variability (see Figure 5). Over the study period, precipitation peaks between July and September, while TWSA peaks between September and November. Ahmed et al. (2014) attributed the observed increase in TWSA to an increase in precipitation in the region caused by warmer Atlantic Ocean temperatures, with the trend in P validated using multiple precipitation sources, including satellite products and rain gauges. While the GPCP data set used here did not show any increase in precipitation, neither did a recent study using rainfall estimates from the Tropical Rainfall Measuring Mission (TRMM) (Ayman and Jin, 2016), so the true cause of this trend remains somewhat unresolved. During the first two years of the GRACE observing period (2003-2004), the basin experienced a downward trend in TWSA. During this time, the $r_l$ values increased at the same time as TWSA decreased towards its minimum value. Then, while TWSA values were recovering, the correlation quickly decreased and became negative. This is similar to what was observed in the Colorado River basin during the dry period. During some wet periods (e.g. July/August 2006, 2007 and 2008), when TWSA increased

towards its highest value, $r_l$ increased and was positive, but then decreased after TWSA peaked. More generally, there seems to be a connection between $r_l$ and the water cycle variations in this region: both high TWSA and low TWSA produced positive correlations. The transitions from positive to negative values make sense considering that when TWSA values approach zero, the observations are more uncertain, as they are affected by noise (i.e. signal to noise ratio). However, the relation might also be influenced by the lag in phase between GRACE observations of TWSA and P-E. Interestingly, there seems to be less inter-degree variability compared to the other basins studied here. This may be related to the simpler water budget in this basin compared to that of the Colorado River and Aral Sea basins, but requires further investigation.

### 4.2.3 Aral Sea basin

The endorheic Aral Sea basin reflected a historical trend of water loss during the study period, most likely caused by anthropogenic consumption related to agricultural activities (Zmijewski and Becker, 2014). Although there were short intense precipitation events during much of the study period (Figure 2), the total annual precipitation showed a negative trend of -31 mm.yr$^{-1}$ from 2003-2008. However, water storage values increased in 2005 as a result of the construction of a dam between the north and south portions of the Aral Sea (Shi et al., 2014). During most of the study period, the $r_l$ values oscillated in a similar way as for the Colorado River basin: that is, a weak connection between high $r_l$ values and increasing or decreasing TWSA, before reaching the local maxima or minima, respectively (see Figure 6). Some examples of this behaviour include June–August 2008 and July-October 2009, before TWSA reaches its lowest value. In general, the $r_l$ values decreased with increasing degree. However, inter-degree variability was more complicated in this basin during several months. Although the Aral Sea basin is predominantly arid, the south-east portion of the basin includes a mixture of warm and cold climates and is where most precipitation occurs. Due to the mismatch in resolution and/or different land cover inputs, the evaporation products may represent these intermixed regions differently. Furthermore, glacial and snowmelt runoff present further complications to the hydrological description. These complications are reflected in the higher inter-degree variability (compared to the other basins). Differences in degree correlation due to the use of the three evaporation products were minimal i.e. no single evaporation product resulted in a significantly higher (or lower) hydrological consistency with precipitation and water storage anomalies.

### 4.2.4 Lake Eyre basin

Another endorheic basin examined here was the Lake Eyre basin, which experienced a marked increase in precipitation during the rainy seasons of 2009-2011 (Figure 2), resulting in an increase in water storage anomalies of about 40 mm.yr$^{-1}$ (calculated from September 2009 to December 2011). The times in which TWSA and P-E were negatively correlated (i.e. negative $r_l$ values) increased during this rainy period (see Figure 7). Total annual precipitation decreased from 2003-2006 (-23 mm.yr$^{-1}$), with a corresponding secular decreasing trend in TWSA of -8.26 mm.yr$^{-1}$. In the same period however, the degree correlations did not reveal any structure or indicate any connection with either P-E or TWSA. A short but intense

precipitation event during the winter of 2006-2007 (Figure 2) did not seem to affect the variations in $r_l$, relative to the earlier years. The $r_l$ variations did show improvements during most of 2008 (when precipitation was low, i.e. P<50mm), particularly with the MOD16 evaporation product (i.e. the retrieval representing the lowest evaporation values). Overall, the $r_l$ values generally decreased with decreasing length scales. Differences in absolute $r_l$ values were visible between the evaporation products, but not in the overall spatial and temporal patterns. More importantly, none of the evaporation products showed a significant (and persistent) advantage in terms of hydrological consistency over the others. Wang et al. (2014) also studied the hydrological consistency of satellite products (TRMM-based P, MOD16-based E and GRACE TWSA) over this basin, as well as other predominantly arid regions of the Australian continent. At the monthly scale, their study also found poor agreement between TWSA and P-E.

**4.3 Applying a phase lag to GRACE data**

For GRACE to identify a water storage increase, the water mass resulting from precipitation needs to accumulate within the catchment beyond a detectable threshold. This accumulation process may take up to several months, during which time the spatially distributed rainfall drains via sub-surface processes or collects in rivers after traveling from different source areas within a basin (Rieser et al., 2010). The apparent lag that GRACE data illustrates relative to faster hydrological processes such as precipitation events has been observed in African basins (Ahmed et al., 2011; Ayman and Jin, 2016) as well as in Australia (Rieser et al., 2010; Wang et al., 2014). The clearest example from amongst the basins studied here is shown in the Niger River basin (Figure 5), where a lag of two months is evident throughout the study period. In other regions, such as the Colorado River basin and Lake Eyre basin, the time needed to detect water storage changes after precipitation events tends to vary, perhaps due to changing spatial and temporal patterns in precipitation as well as geomorphological characteristics (Ahmed et al., 2011; Wang et al., 2014). Because of their large extent and geographical features, the Colorado River and Aral Sea basins include regions where snow storage plays an important role as a source of delayed runoff. The combination of snowmelt, groundwater flow and other sources of delayed flow are defined as baseflow (Beck et al., 2013).

To examine this temporal component, at least in a simplified manner, a lag of one, two and three months was considered for all basins and assumed to remain constant throughout the study period. In terms of changes to the degree correlation, for the Niger River basin it was clear that a two months lag produced an improved temporal match between TWSA and P-E. For the other basins however, due to the changing dynamics in precipitation and TWSA, a temporal match could not be satisfied at all times by using an arbitrarily constant lag in GRACE. Regardless, it was found that a persistent lag of two months provided a better fit compared to all alternatives (including zero lag). Beck et al. (2013) developed global estimates of the Base Flow Index (BFI): a measure of the ratio of the long-term baseflow to the total runoff, using a large global data set of runoff and a regionalization procedure to transfer these and other characteristics of runoff from gauged to ungauged basins. Since we did not model any of the physical processes contributing to baseflow, the BFI was examined to assist in explaining part of the delay in observed water storage changes relative to the P-E term (although not dynamically, since the index is a

long-term average in time). The spatial average of the BFI in the four study basins is, not surprisingly, within the same range: between 0.4 and 0.6. This is not unexpected, as various climate characteristics were used as predictors of BFI. Indeed, the fact that they are similar is in agreement with our finding of similar GRACE lag times among the study basins. Further investigation is required to determine the nature of the elements affecting the lag in water storage, not limited to those found

in baseflow.

Figure 8 presents a statistical summary of the mean degree correlation values over the study period, comparing the original analysis and using a constant lag of two months. The results are presented as boxplots, where the median is indicated as a bold black line inside a box confined by the first and third quartiles (bottom and top of the box). The whiskers below and

above the first and third quartiles show a threshold of 1.5 times the inter-quartile range (IQR), defining a number of outliers outside this range. As already noted, the Niger River basin showed a significant improvement in $r_l$ after considering the delay, not only in terms of the median $r_l$ value, but also in terms of the variability in the results (i.e. a smaller IQR). This outcome was similar irrespective of the evaporation product used. For the Colorado River basin, the degree correlations did improve when using the CSIRO-PML and GLEAM products (median improved from 0.17 to 0.67, and from -0.01 to 0.64,

respectively) but to a lesser extent for the MOD16 product (-0.03 to 0.29). The IQR was also reduced significantly with the CSIRO-PML product, moderately reduced with GLEAM, and did not change with the MOD16 product. The degree correlation in the Aral Sea basin also benefited from an imposed lag in GRACE data, although there remained considerable variability in the results. The Lake Eyre basin showed only a marginal increase in the amplitude of $r_l$ and a minor reduction in the temporal variability (-0.06 to 0.14, 0.08 to 0.20 and 0.13 to 0.29 with CSIRO-PML, GLEAM and MOD16

respectively).

## 5 Discussion

The development of methods and sensors to retrieve the various components of the water cycle has for the most part been undertaken independently of any evaluation against interrelated processes (see McCabe et al., 2008 and Brocca et al., 2014 for some examples of complementary retrieval). Large-scale retrievals of hydrological variables such as evaporation, soil

moisture and rainfall products do not come with well-defined accuracy metrics, let alone uncertainty bounds. This lack of any well-defined error structure associated with individual products complicates the task of product assessment. As such, the question of how to evaluate large-scale datasets remains an outstanding one. This is especially important in the context of global-scale products. While a number of global evaporation (and precipitation) evaluation papers have been published, none seek to identify consistency with related hydrological variables, and focus instead on comparisons against traditional point-

scale or tower-based techniques (McCabe et al., 2016; Michel et al., 2016). Given the spatial mismatch between ground observations (and the lack of continuous large-scale coverage of in situ data in remote regions), it is perhaps inappropriate to evaluate these large-scale products in such a manner. Determining whether individual products are at the least consistent

with each other (i.e. they reflect hydrological expectation) is a needed first step in product assessment. The motivation behind this study was to take a step back and determine whether a first order hydrological assessment could be achieved. Rather than comparing the uncertainties between the evaporation products and the other hydrological components (which are poorly defined), we attempt to distinguish between the different evaporation products relative to their consistency with precipitation and terrestrial water storage. That is, are observed changes or patterns in the evaporation datasets reflected in these other hydrological variables? We explore this approach precisely because of the challenges in quantifying uncertainty based upon traditional in situ methods. As is discussed below, the challenge on how to do this remains, raising some important questions on both product quality and also the techniques we use to evaluate global products.

**5.1 Challenges to implementing hydrological consistency**

For some regions, especially those where simpler and more defined water cycle behaviour dominates, it is reasonable to expect that significant and consistent inter-product agreement between hydrological components should be observable. To explore this idea, our study focused on basins where such a simplified water budget, consisting of water storage anomalies as a function of precipitation and evaporative fluxes, might be expected to predominate. The aim was to reduce the influence of complicating variables such as snow, vegetation changes, large precipitation and streamflow contributions and other hydrological processes from the analysis. The assumption was that arid and semi-arid regions would best fit this profile. The role of the degree correlation was to evaluate the spatial agreement between the hydrological components, assuming that any non-closure errors due to unmodeled outflow components (e.g. long-term baseflow or minimal surface runoff) would not affect this measure. However, other sources of errors that directly affect evaporation estimates, such as the choice of algorithm, implicit model assumptions, choice of parameterizations and an incorrect representation of the land cover, can directly impact the degree correlation measure. Given the relationship between size and retrieval accuracy as relates to GRACE data, obtaining a geographical distribution of basins that could satisfy this simplified water budget assumption was non-trivial. Restrictions related to basin size affect the study in two conflicting ways. On the one hand, a large basin will inevitably present complications related to heterogeneity (including in climate zones, as was the case for the Colorado River basin and Aral Sea basin) and also be more likely to contain areas affected by anthropogenic activities, such as irrigation, land cover changes, building of dams and reservoirs, etc. On the other hand, a small catchment size is more difficult to evaluate with this consistency approach, given the coarse resolution of (most) of the global products used here, but especially the GRACE data. The spatial resolution of GRACE data is further limited by the use of filters to remove errors. Considering these restrictions, a compromise in the selection of study basins was required to allow for at least a narrow range of length scales (500-800 km) to be evaluated.

In the end, our study consisted of four major globally distributed river basins, including two endorheic systems. Although having mostly an arid climate in terms of Köppen classification, both the Colorado River and Aral Sea basins include regions with the presence of snow and snowmelt-dominated runoff. While snow storage itself is not a problem, since GRACE

detects changes in storage irrespective of their nature (snow, groundwater, soil moisture, etc), snowmelt may contribute to delayed changes in storage that can affect gravity results. Likewise, evaporation models generally have a difficult time adequately estimating sublimation. However, the inclusion of these basins was considered important in order to test the hydrological consistency concept in regions that deviated from the ideal assumption. Indeed, the influence that snowmelt and other potential sources of lag in the system have is poorly defined and forms part of the motivation to explore the inclusion of a lag response in the GRACE data (see Section 4.3).

Apart from the issues of spatial scale, the use of satellite-based hydrological data presents additional challenges and sources of uncertainty to any consistency-based assessment. For instance, because GRACE data is smoothed to remove errors in small-scale terms (i.e. truncation of the spherical harmonic coefficients), the gravity signal contains contamination from outside of the studied basins (leakage) and represents a potential source of uncertainty in areas neighbouring high amplitude signals (particularly if they are out of phase with the study basin) and the ocean. Although the LSM-based scaling factor, which is static in time, has been used to correct for bias (e.g. signal reduction) and leakage contamination, dynamic changes in water storage trends outside the basin might still contaminate the signal (Long et al., 2015). In addition, the temporal lag in terrestrial storage response, as documented in previous studies (Rieser et al., 2010; Ahmed et al., 2011; Wang et al., 2014; Ayman and Jin, 2016) and observed in our analysis, represents an important source of potential error (see Sections 4.3 and 5.2). Product errors are also evident in the precipitation and evaporation data sets. Global rainfall retrievals have well recognised limitations, including the detection of both high and low intensity events (Hou et al., 2014), the discrimination of cloud free and cloud precipitation scenes, as well as the sensitivity to parameters in the forward model of radiative transfer over different sensors (Stephens and Kummerow, 2007). In terms of evaporation, uncertainties related to algorithm choice, input data variability and process parameterizations all complicate the accurate estimation of terrestrial fluxes (Ershadi et al. 2015). McCabe et al. 2016 present a thorough description of accuracy issues related to global products. However, it is not the intent of this work to explore these product uncertainty issues in detail. Determining whether or not and understanding how much these sources of product uncertainty affect hydrological consistency studies is an important area requiring further investigation. What is clear from this analysis is that there is still some way to go in terms of being able to confidently assert that any single global product outperforms any other: at least in terms of its inter-product consistency.

## 5.2 Temporal lag in terrestrial storage response

In exploring the relationship between GRACE water storage changes and precipitation and evaporation data, it was evident that water storage anomalies peaked at a significantly later time than the corresponding P-E values. One possibility for this apparent lag in GRACE data is that, due to the inability of GRACE to detect small-scale changes in the gravity field (a rough estimate of GRACE accuracy averaged over the entire Earth is 20 mm.month$^{-1}$; Wahr and Velicogna, 2006), the corresponding mass is not detected until a sufficient amount has accumulated within the catchment via natural drainage

processes (Rieser et al., 2010; Ahmed et al., 2011). The intensity and duration of the precipitation events, antecedent soil moisture condition, as well as the hydrogeological and geomorphological characteristics of the basin all influence the accumulation and detection time. A simple way to account for this phenomenon was to apply a constant phase lag to GRACE data across the whole study period. Doing this improved the behaviour of degree correlation, not only in time (less

variability in the results), but also increased the value of $r_l$ as well. This was particularly evident in the Niger River basin, which was expected due to the well-defined seasonal behaviour of its hydrological cycle throughout the study period, and to a lesser extent in the Colorado River basin and Aral Sea basin, where changing trends in the seasonal patterns of precipitation made it more challenging to apply this simple correction. In the Lake Eyre basin, applying a lag to GRACE data did not seem to have an effect on the degree correlation. Clearly, this is a simple approach at accounting for delayed

flow contributions to the catchment, and further understanding the implications and physical rationale behind the attribution of this lag is required.

## 5.3 Discriminating between satellite evaporation products

One motivating aspect of this work was to explore whether differences in available global evaporation products (i.e. satellite-based evaporation models forced with global input data) impacted the results of the consistency analysis i.e. could we

identify better agreement between water storage anomalies and P-E in any particular evaporation product? While the analysed products covered a wide range of resolutions (0.05°–0.25°), the effective resolution in the analysis was ultimately determined by the truncation degree ($l_{max}$) of the spherical harmonic transformation. Even after accounting for this, absolute differences were evident from a qualitative basin-scale analysis (Figure 2). Results indicated that MOD16 underestimated evaporation when compared to CSIRO-PML and GLEAM, even though both the CSIRO-PML and MOD16 products are

based on the Penman−Monteith equation. Several recent studies (McCabe et al., 2016; Michel et al., 2016; Miralles et al. 2016) also suggest that the MOD16 product (or variants using the PM-Mu approach) underestimate evaporation when compared to other products (including GLEAM), and that most products show large discrepancies in reproducing results during periods of water stress. Ershadi et al. (2015) demonstrated that the parameterization of aerodynamic and surface resistances were critical controls on evaporation through both soil and vegetation. Furthermore, both GLEAM and CSIRO-

PML include dynamic constraints on evaporation (stress module and soil moisture assimilation in GLEAM; dynamic ratio of actual to potential evaporation at the soil surface in CSIRO-PML) that are critical in arid regions due to hydrological and plant physiological stresses and the subsequent importance of soil evaporation. Whether these differences in model parameterization are the sole cause of the apparent underestimation by MOD16 remains to be investigated. However, these differences in absolute values did not affect the overall results of degree correlation dramatically, i.e. we could not identify a

significant advantage or disadvantage in terms of hydrological consistency.

As the focus of the study was to discriminate between evaporation products, the question on whether the choice of precipitation product affected the hydrological consistency analysis was somewhat beyond the scope of this work. However,

a preliminary analysis was undertaken by replacing the GPCP precipitation data with another data set and reproducing the analysis. To do this, we processed the Precipitation Estimation from Remote Sensing Information using Artificial Neural Network (PERSIANN) product, which uses an artificial neural network to approximate spatiotemporal non-linear relationships between physical variables and remotely sensed signals (Hsu et al., 1997). PERSIANN uses data from the long

wave infrared imager onboard the Geostationary Operational Environmental Satellite (GOES) as well as from the Tropical Measurement Mission (TRMM) microwave imager (TMI). As shown in Figure S1, the results of this new analysis did not reveal any significant difference when compared to those based on the GPCP analysis. Figure S1 shows the average degree correlation statistics per study region and evaporation product, with and without the inclusion of a lag in GRACE data.

Evaluating global evaporation products remains an outstanding challenge. The purpose of implementing a hydrological consistency approach was to explore the evaluation of evaporative fluxes by comparing the spatial patterns between precipitation and changes in water storage. If such an approach could be shown to perform well in a relatively simple hydrological system, the potential for broader-scale application in regions with more complex behaviour would be the next logical step. However, the study showed that even in these relatively simple basins, it was not possible to demonstrate a

consistent hydrological agreement between independent observations. Improvements in satellite-based evaporation products are likely to be delivered through advances in algorithm development, increases in the observable resolution and also via the development of multi-product ensembles (with weighting based on validation analyses and uncertainty assessments). The prospects for improved precipitation monitoring is also promising given the Global Precipitation Measurement mission, which will allow for a more accurate representation of light rains: a challenge that has been a limitation in other precipitation

products, including the GPCP (Huffman et al., 2001). Likewise, the next generation gravity missions (GRACE follow on and GRACE II) with the incorporation of improved sensor design (Christophe et al., 2015) are anticipated to provide more accurate estimates of the water storage anomalies, albeit with no significant increase in resolution.

**6 Conclusions**

Given the inherent challenges in validating satellite-based products via the use of ground-based observations, a key

motivation behind this study was to examine the capacity of independent observations of the water cycle to reflect some form of hydrological consistency. To do this, the study focused on regions where it would be most expected to observe such responses: arid and semi-arid regions with a simplified water budget, consisting primarily of precipitation and evaporation, and assuming a minimal runoff and other long-term outflow components. Unfortunately, it was determined that even in these simple environments, hydrological consistency was difficult to obtain. While there are times and locations at which some

consistency was observed, there were a greater number for when it was not. The lack of any persistent behaviour is problematic, both in the attempt at independently evaluating remote sensing data and also in any effort to discriminate between individual products. Although there were significant and known differences in evaporation estimates, especially

with the MOD16 product in the Colorado river and Aral Sea basins, these differences, whether caused by model parameterizations or by input data, did not seem to play a significant role in the overall level of hydrological consistency.

While not envisioned as providing a comprehensive tool for product evaluation, the approach did help to reveal some interesting spatial and temporal patterns between the studied hydrological variables. In general, the correlation between the satellite products was higher with smaller degrees, or larger spatial scales. In simple water cycle systems such as in the Niger River basin, the correlation followed cyclical patterns along with the water storage anomalies i.e. correlation increased together with water storage anomalies up to the point where these peaked, where it then decreased to the point where these were minimal. A similar pattern but in terms of negative correlations and negative anomalies was also observed. This indicates that, at the least, the correlations are not random, but roughly follow the cyclical hydrological variations within the basin. It is also quite reasonable to expect low agreement when fluxes and/or water storage anomalies are minimal, explaining some of the cyclical nature in the correlation. A lag between GRACE and precipitation data was also considered to account for delayed sources of water storage changes. It was shown that imposing even a simple correction (i.e. a constant phase shift to GRACE data) greatly improved the agreement, both in average degree correlation and variability of the results in time. Implementing techniques to better account for these delayed sources of outflow could prove highly beneficial for the analysis of hydrological consistency.

The lack of persistent agreement in some of the studied basins may be explained in part by the added complexities that limit the validity of the assumption of a simple water cycle i.e. snow melt runoff, complex geomorphology or hydrogeology, changing patterns of precipitation as well as anthropogenic influences on the water system. Other limitations to exploiting the hydrological consistency approach include the many challenges that still exist in the large-scale retrieval of precipitation, evaporation and GRACE data, all of which complicate a thorough interpretation of product uncertainty. Despite these challenges, the expectation is that retrievals of global and regional products will inevitably improve with advances in resolution, process understanding and forcing data accuracy. In concert with such product improvements, the way in which we evaluate remotely sensed variables should also evolve beyond the relatively simplistic comparisons against in-situ data that form the basis of most current assessments. Such a strategy would include evaluation against related hydrological variables, reflecting the underlying rationale of hydrological consistency and hydrological closure studies. Only by implementing a more comprehensive evaluation framework into our assessment schemes will greater confidence in component retrievals be realized.

**Acknowledgments.** Research reported in this publication was supported by the King Abdullah University of Science and Technology (KAUST).

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

| Product name | Spatial resolution | Time span | Reference |
|---|---|---|---|
| *Evaporation* | | | |
| MOD16 (A2) | 0.05° | 2000 - 2013 | Mu et al. (2011) |
| CSIRO-PML | 0.25° | 1981 - 2011 | Zhang et al. (2012) |
| GLEAM (v2A) | 0.25° | 1980 - 2011 | Miralles et al. (2011a,b) |
| *Water storage* | | | |
| GRACE (CSR RL05 GSM) | 333 km ($l_{max}$ = 60) | 2003 - present | Tapley et al. (2004) |
| *Precipitation* | | | |
| GPCP (1DD v1.2) | 1° | 1996 - present | Huffman et al. (2001) |

Table 1. Description of the satellite products used in this study. The temporal resolution is daily except for MOD16 (8-daily) and GRACE (monthly). The original MOD16 product is available at 1 km resolution in the sinusoidal projection. In this study, the product was reprojected onto a 0.05° regular grid using the MODIS Reprojection Tool (MRT).

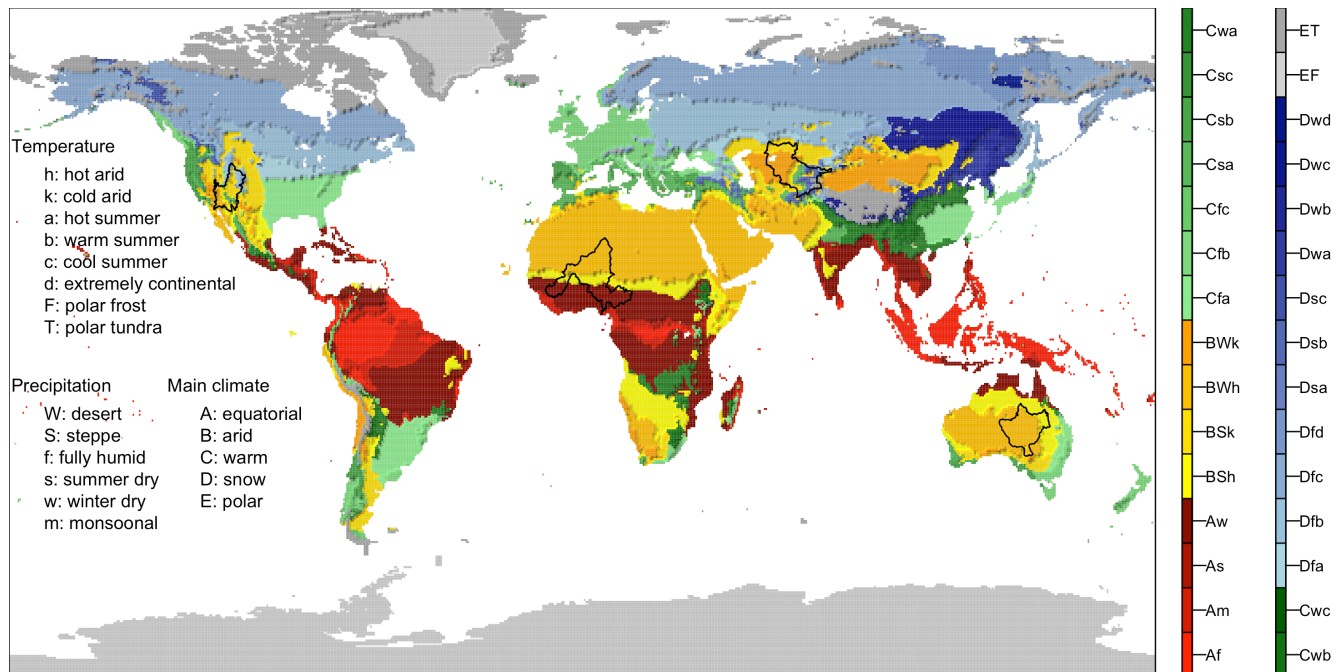

Figure 1: Selected study basins used within the analysis. Criteria for the selection of basins included: predominantly arid climate (more than 50% areal coverage with any of the arid Köppen climates: BWk, BWh, BSk or BSh), size, geographical location and amplitude and trends in the water storage variations.

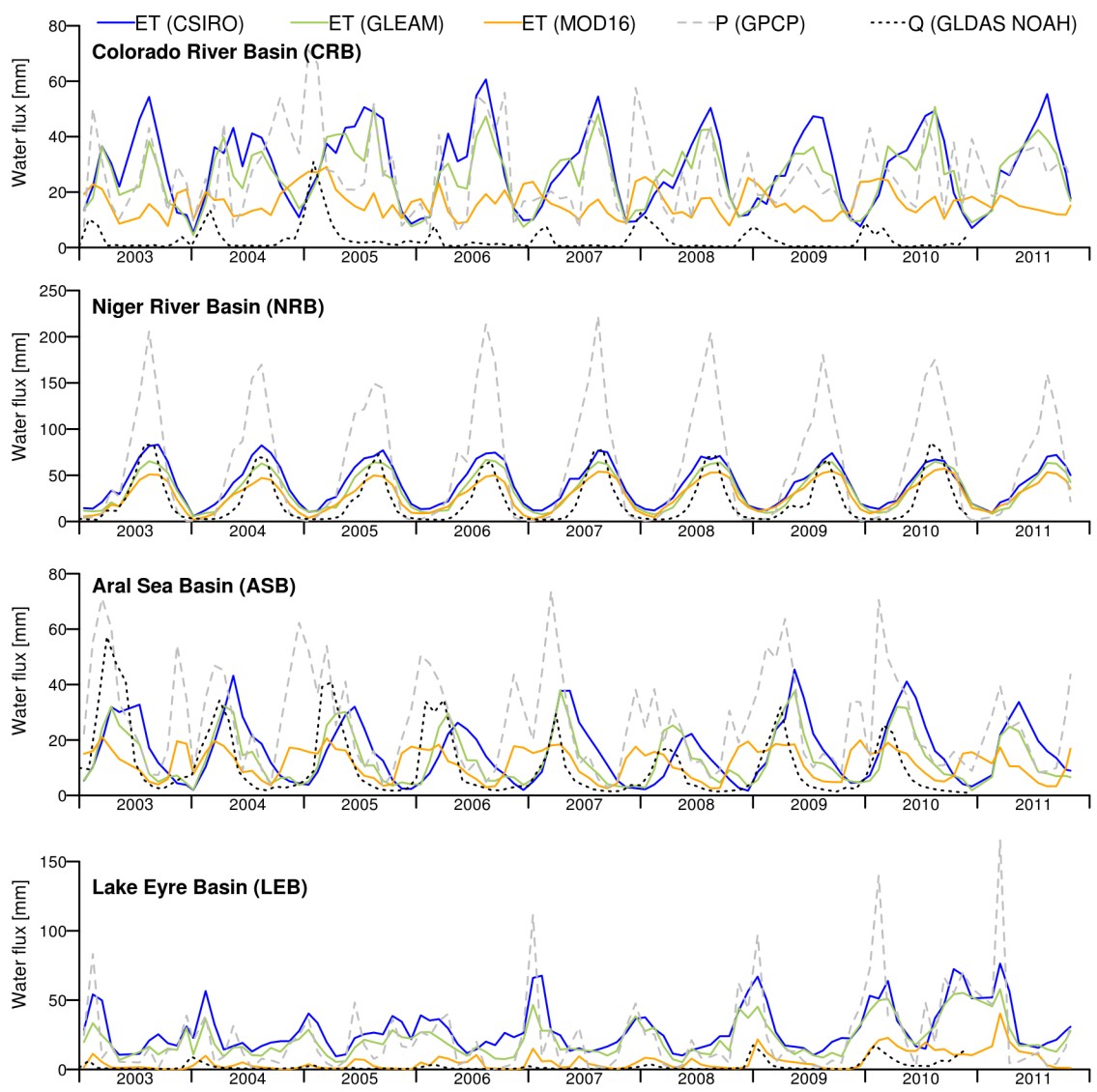

Figure 2. Average P, E and Q fluxes within the four study basins for the period 2003-2011. Three evaporation (E) data sets were used in this study, including CSIRO-PML, MOD16 and GLEAM (see Table 1 for details).

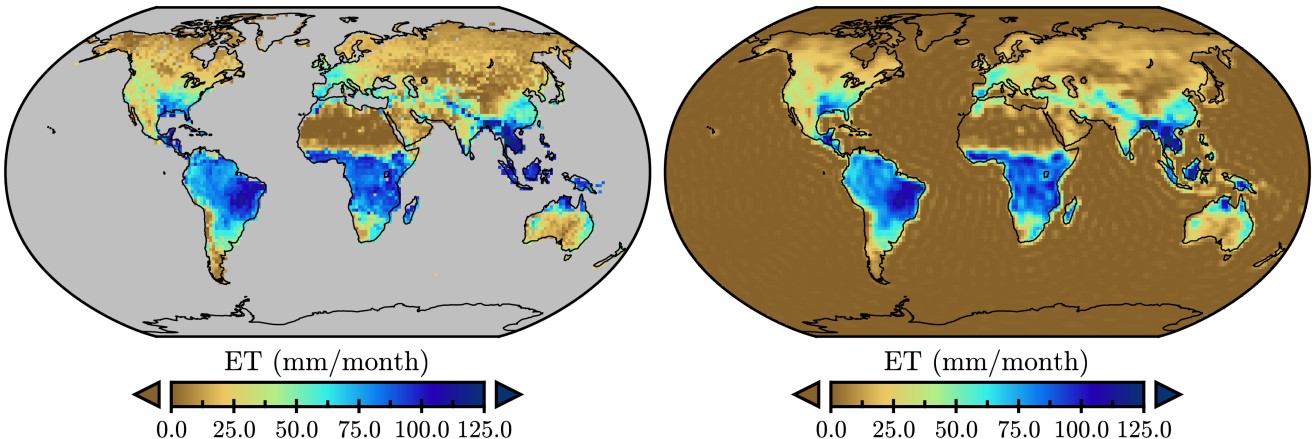

Figure 3. Left: CSIRO-PML monthly evaporation (E) for April 2003. Right: the same data set after spherical harmonic analysis and synthesis of the evaporation data. Missing data are set to zero. The data appears smoothed because it is an approximation (equation 1) limited by $l_{max}$. There are also other effects such as ringing that are visible in regions where the data is close to zero.

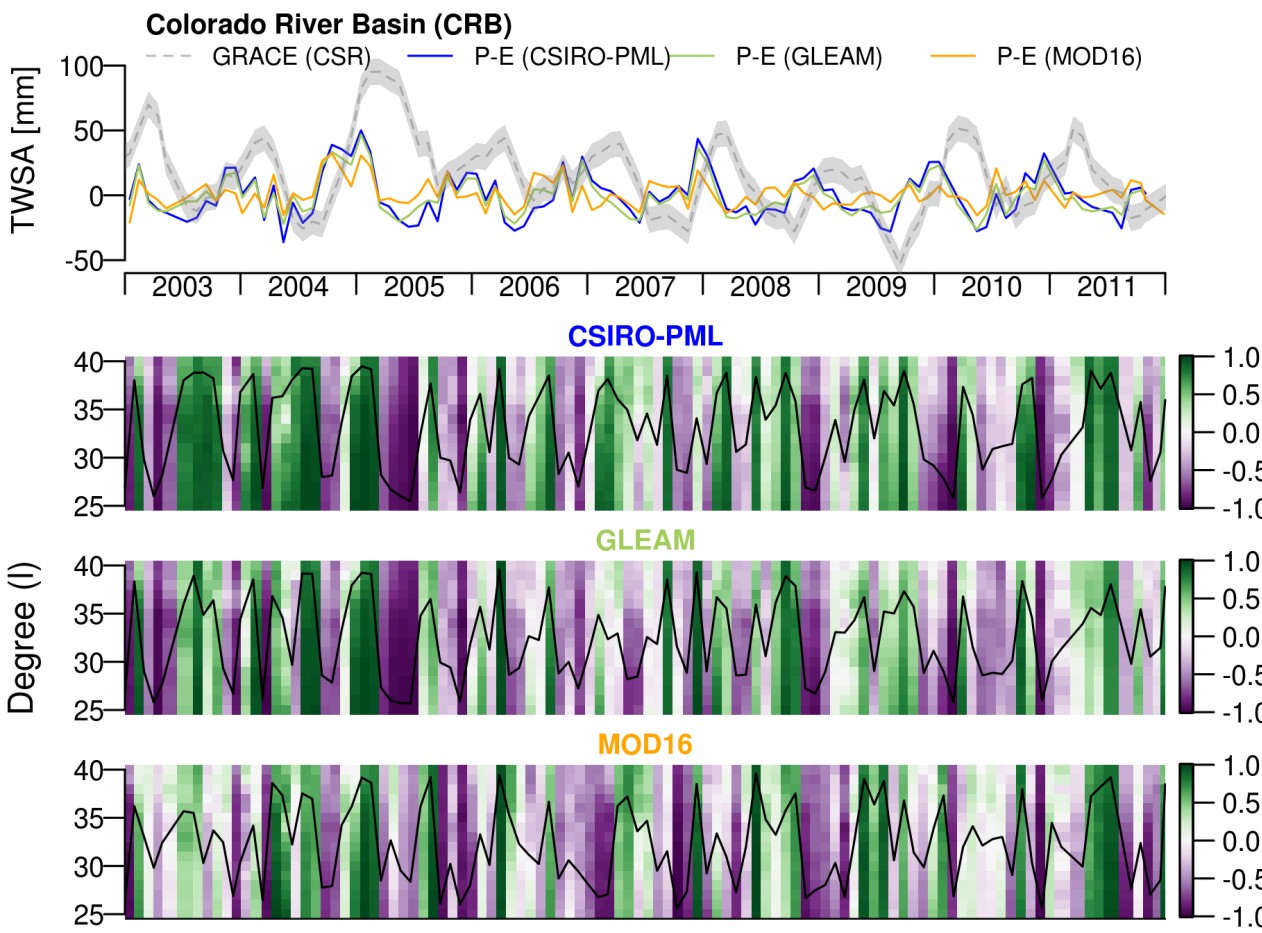

Figure 4. Top: anomalies of the terrestrial water storage (TWSA) observed by GRACE (with 20 mm uncertainty bounds) and P-E using three global evaporation products over the Colorado river basin. Below: varying degree correlation measure ($r_l$) with time and degree (from 25 to 40) using the three global evaporation products. The average $r_l$ is shown as a time series (black line). The degree correlation measure can range from -1 to 1 as shown in the color scale on the secondary axis.

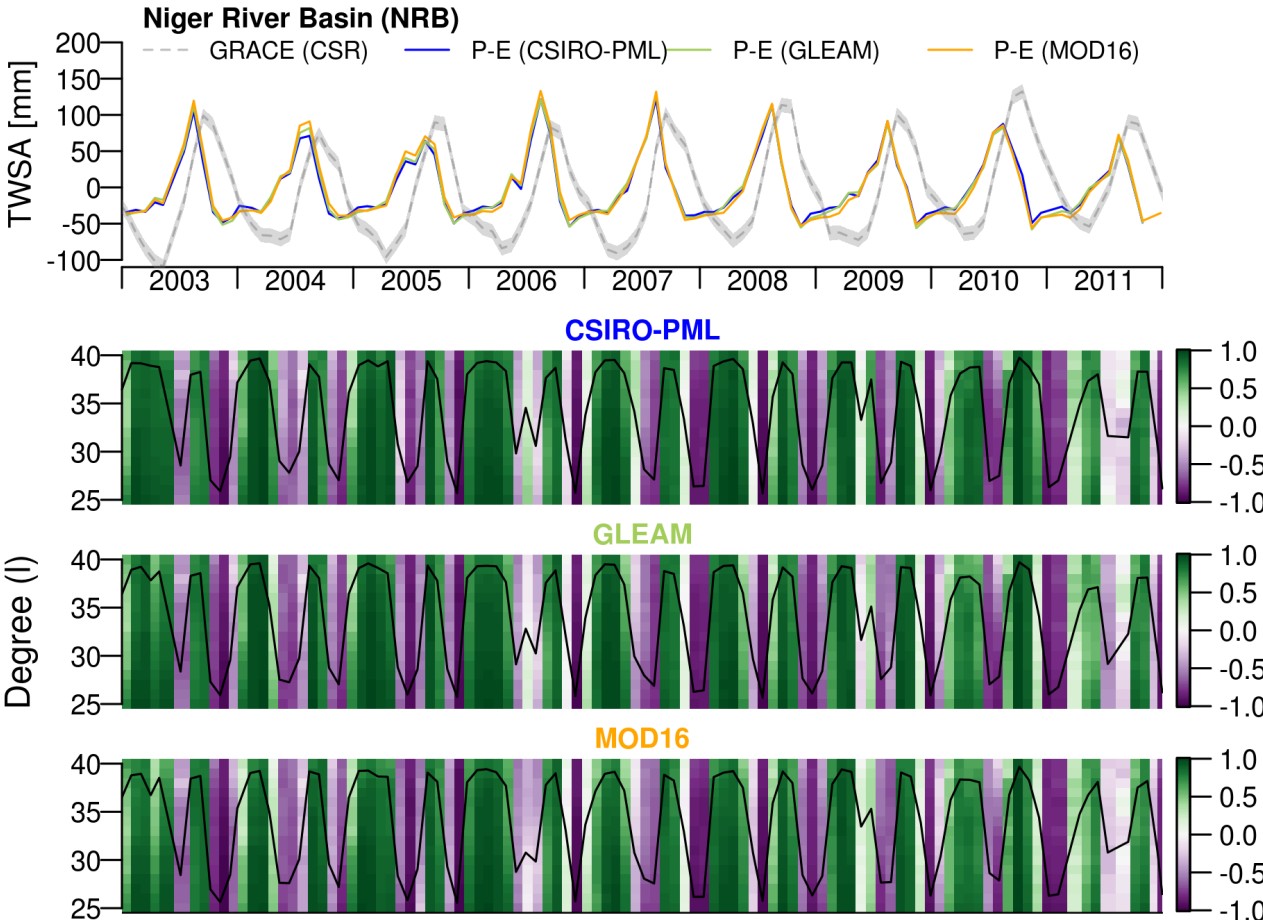

Figure 5. Top: anomalies of the terrestrial water storage (TWSA) observed by GRACE (with 20 mm uncertainty bounds) and P-E using three global evaporation products over the Niger river basin. Below: varying degree correlation measure ($r_l$) with time and degree (from 25 to 40) using the three global evaporation products. The average $r_l$ is shown as a time series (black line). The degree correlation measure can range from -1 to 1 as shown in the color scale on the secondary axis.

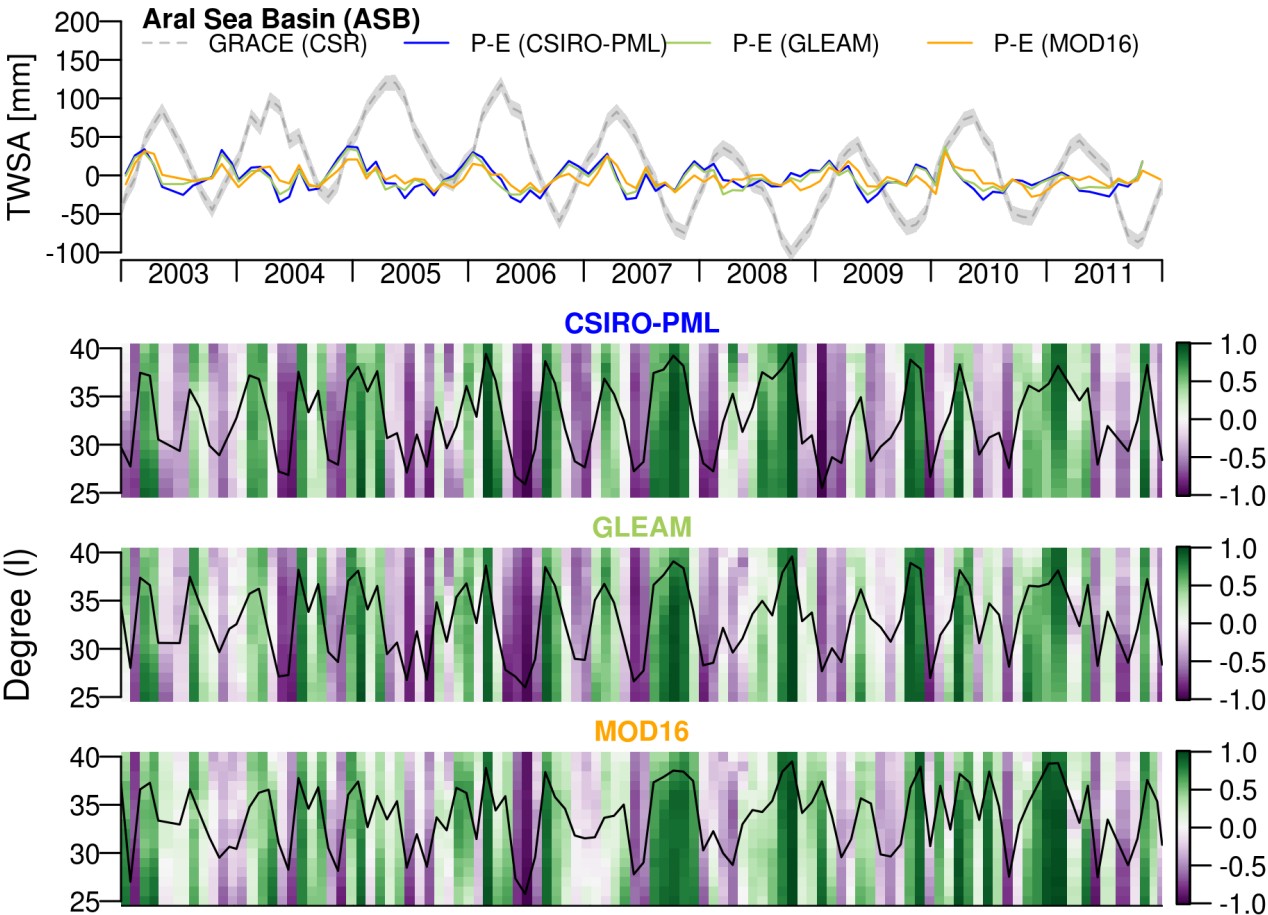

Figure 6. Top: anomalies of the terrestrial water storage (TWSA) observed by GRACE (with 20 mm uncertainty bounds) and P-E using three global evaporation products over the Aral basin. Below: varying degree correlation measure ($r_l$) with time and degree (from 25 to 40) using the three global evaporation products. The average $r_l$ is shown as a time series (black line).

5    The degree correlation measure can range from -1 to 1 as shown in the color scale on the secondary axis.

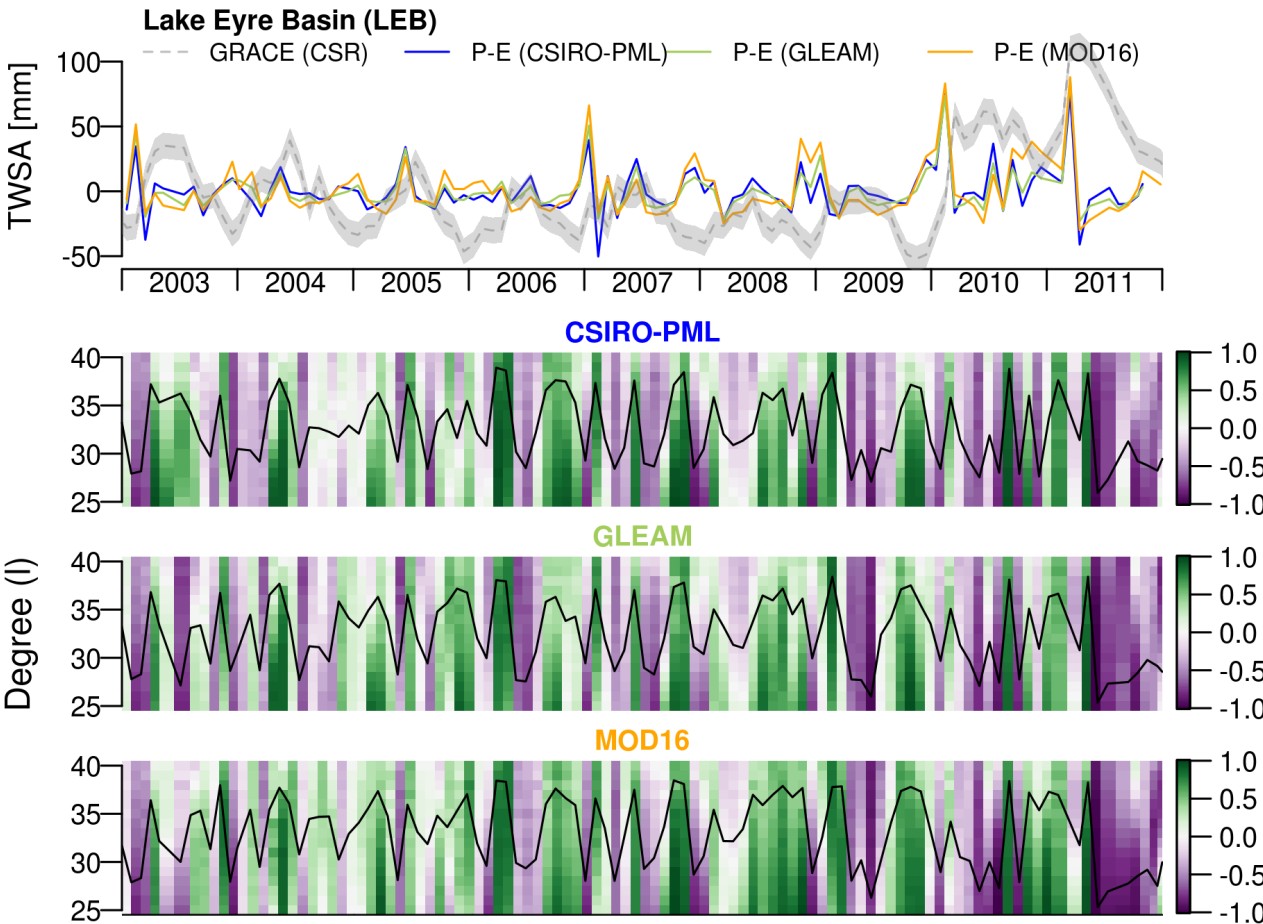

Figure 7. Top: anomalies of the terrestrial water storage (TWSA) observed by GRACE (with 20 mm uncertainty bounds) and P-E using three global evaporation products over the Lake Eyre basin. Below: varying degree correlation measure ($r_l$) with time and degree (from 25 to 40) using the three global evaporation products. The average $r_l$ is shown as a time series (black line). The degree correlation measure can range from -1 to 1 as shown in the color scale on the secondary axis.

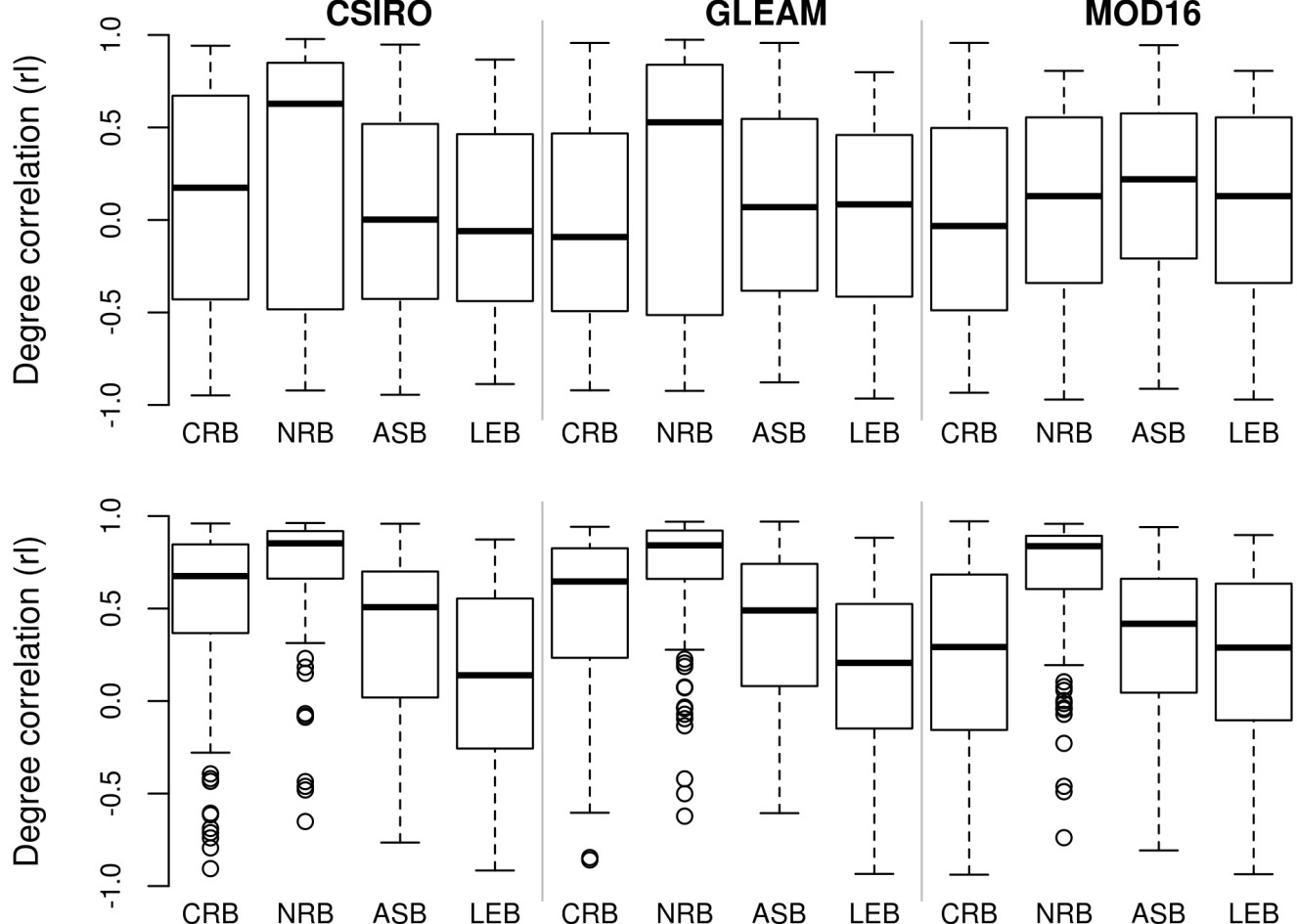

Figure 8. Top: average degree correlation statistics per study region and evaporation product. Bottom: GRACE data were shifted by two months to match the phase with P-E anomalies. The boxplots show the first, second (median) and third quartiles. Outliers, defined as data outside the 1.5 inter-quartile range (IQR) whiskers below or above the first and third quartiles are shown as circles.