# Peer review of "Evaluating the hydrological consistency of evaporation products using satellite-based gravity and rainfall data"

_Hydrology and Earth System Sciences, 2016_

## Short Comment (SC2) · 29 Jun 2016

We initially considered the inclusion of more satellite-based evaporation models, including PT-JPL and ALEXI. However, we limited the study to products with already defined data inputs for global estimation of ET.

---

## Referee Comment (RC1) · R.S. Westerhoff (Referee) · 11 Jul 2016

Dear authors,

This paper describes an evaluation method to explore correlation of satellite-derived water budget components P-ET and Terrestrial Water Storage. It uses a spherical harmonic analysis to analyse correlation and differences between mulitple time series of P-ET and GRACE Terrestrial Water Storage. The method uses consistent GPCP input on P, and three different ET methods. The method results cannot explain differences of the analyses in the three large-scale basin studies.

This study uses a novel approach to estimate differences between two time series. The fact that this analyses does not lead (yet) to valuable results, is no reason to disqualify it in any way. Therefore, it makes this paper a valid and potentially useful

addition to the journal. However, the descriptions and approach are, in my opinion, not fully crystallised and require more research. Some of the descriptions are also rather confusing and need a better structuring (see further comments below). I get the feeling that recommendations described in this work should have actually be part of the paper.

The whole term 'hydrological consistency', which is everywhere in the paper, e.g. the title, is not very clearly explained. Could you consider a better way to describe what you want to research? For example, 'the ability to balance the water budget' or something containing the words 'hydrological closure'?

In terms of concise descriptions: in my opinion, this study knows too many research questions and too little answers. It needs serious work on structure, correct descriptions and conciseness. The authors describe several paragraphs saying: 'the objective of this work', or 'a secondary objsctive of this work was..', or 'one key motivation of the study..'. Or they reword their aims in questions, such as: 'how accurate are the hydrological components derived from satellite observations?' or 'is hydrological consistency acheved with a particular product....?, etc. To me, that makes it confusing. Can you please clearly state the objectives and motivation at first, and then relate back to exactly those? That would make this work better readable. Furthermore, the many different aims and questions also makes you wonder about the conciseness and correctness of this study. For example, the questions "how accurate are the hydrological components derived from satellite observations?" cannot be answered with the results of this study. If all these research objectives and questions would be compiled into only two research questions, I think the paper would be more understandable.

It is unclear what the added value of the spherical component analysis is. After all, correlations between water budget components can also be made in a different way, and the spherical component analyses does not lead to any new insights. Moreover, the lag between some data make the results of the analysis method less obvious. The method does not lead to any quantifications of if the water budget is in balance, or by how much it is off (for example, in percentages of the total water budget). That makes

this study for me hard to judge: it evaluates water budgets nut this does not lead to any new insights (unless you describe that part better), and cannot answer obvious explanations for imbalance of a water budget.

It is unclear why the focus is put on the ET component. You should pitch that better. After all, any uncertainty of the P component would result in larger uncertainty. I think the discussion and conclusion of this study need to point out that comparisons in catchment study need to be undertaken using the regional information on hydro(geo)logy and ground estimates of P, ET and streamflow. My guess is that you want to say that, despite being the second-largest component of the water budget, ET is most uncertain?

The word groundwater is mentioned in the description of GRACE data. However, it is surprising that the word groundwater is not mentioned when discussing the phase lag to GRACE data, nor the separation of the P-ET(=Q) into streamflow and groundwater flow. One could for example compare global data of baseflow (BFI) and look if these compare to the differences in lag. See for example the wonderful work of Beck et al. (2013): Beck, H. E., A. I. J. M. van Dijk, D. G. Miralles, R. A. M. de Jeu, L. A. Bruijnzeel, T. R. McVicar, and J. Schellekens (2013), Global patterns in base flow index and recession based on streamflow observations from 3394 catchments, Water Resour. Res., 49, 7843–7863, doi:10.1002/2013WR013918.

It is also surprising that nothing is said on snow storage in the discussion on P-ET.

Furthermore, the description of input data is not clear. Espexially the ET part should contain what part of the data we use. For example, do we use PET from MOD16 or AET? Also, MOD16 contains an unclear description that first suggests it is a Penman-Monteith method, and then suggests it is Priestley-Taylor. Furthermore, the spatial detail in Table 1 on MOD16 is not correct to my knowledge and resolution should be 1km (at least if the data from Mu et al from http://www.ntsg.umt.edu/project/mod16 is used). See more detailed text comments below.

Methods and results are in my opinion too intertwined, and should be separated more. For example, in the results, there are some descriptions of the basins. For example, descriptions of the CRB, ASB and LEB basins contain texts that should in my opinion be put into 2.5 Study Regions.

I am not an expert in spherical harmonic analyses, so I cannot judge on the correctness of the method. However, I think it would make the text a bit clearer if the term 'degree' (l) would be explained a bit more in detail, so non-experts in spherical harmonics would also understand. In that way, your plots 4-7 would be easier to understand.

A weakness of this paper is that it does not incorporate uncertainty. After all, estimates of ET using ground observations already contain quite a bit of uncertainty (see e.g. Westerhoff, 2015, with work on Uncertainty of Penman and Penman-Monteith estimates). Another weakness of this paper is that it is not clear what specific ET has been used (PET, AET, P-T, P-M). A quick fix could be a better description of the input data. However, a discussion of different methods used (e.g. P-T or P-M, or PET or AET etc) should be clear on pointing out differences in methods. It would be unfair to judge that satellite data does is not hydrologically consistent, if the ground-estimates already are so uncertain and if the different methods have caused this uncertainty. It is also unfair to put 'the blaim' on ET, because P is a much more likely candidate for this uncertainty. You need to describe better why ET is so important in this analyses. My guess is that you want to say that, despite being the second-largest component of the water budget, ET is most uncertain. Can you derive uncertainties of all estimates? This study would be much stronger if it could quantify what absolute values of inconsistencies we are talking about.

If the document would be structured better, I think it would improve the quality and probably I would understand better what the exact goal is of this paper. Reducing the number of research questions/objectives would probably clarify why this paper should be published. Therefore, I recommend a major revision.

[Figure]

More detailed comments:

Page 1, Line 10: 'regional-to-global', I think it should be just 'regional to global'. Check throughout the document.

Page 2, Line 14: 'in-situ' should be 'in situ'. Check throughout document.

Page 2, Line 16: insert space between 'low,independent'.

Page 2, Line 18: 'land-surface' should be 'land surface'

Page 3, Line 16: I think you should mention that ground-observed estimates of ET also contain uncertainty.

Page 5, Line 1-11: Are you using PET, AET, PLE or LE?

Page 5, Line 10. You should use the correct definition. Mu et al use a P-T estimate for plant transpiration. Consider removing this sentence, since it is confusing and it now reads as if Mu et al use a P-M and a P-T approach for ET.

Page 5, Line 13-19: what are you using? AET?

Page 5, Line 20-26: please mention that GLEAM is AET.

Page 6, Line 9-15: starting with mentioning 'although not used for...' confuses the reader. Please start by saying what it is, what is has been used for in this study, and what is has not been used for.

Page 6, section 2.5. Some of the texts of methodology that described the basins, should be in here.

Page 7-8: Please describe what the term 'degrees' is and what it means. Since you rely on it quite heavily in your figures 4-8.

Page 9, Line 20-25. This is quite a statement. You assume that clear challenges are needed to reach hydrological closure using satellite data. That is already a paper conclusion. I think you should include something here that mentions that this is nothing

new (i.e. closing a water budget has always been a challenge, with or without satellite data) and that the other uncertainty of ground-observed estimates and models of ET is still under-explored.

Page 13, Line 6: 'interrelated' should be 'inter-related'

Page 13 - Page 15, Disucssions. I had a really hard time reading through the discussions. This should be better structured in topics.

Page 22, Table 1. The resolution of MOD16 is 1km on the sinusoidal MODIS grid if you take Mu et al data. It is unclear what the 0.05 arc-degree is doing here.

Page 24: Figure 2: it would have made more sense to compare P-ET with Q in these studies. The color scale in the figures could use a legend (is this rl?).

---

## Referee Comment (RC2) · Anonymous Referee #2 · 14 Jul 2016

The paper is quite well written and the topic is important. However, there are unfortunately some serious problems with the paper:

1. The study focuses on four "relatively simple" catchments. But what's so simple about these catchments? The runoff fraction is definitely not negligible (see Figure 2). The snowfall fraction in some of them is quite high (e.g., precipitation in the Aral Sea and Colorado catchments is composed of about 15% snowfall). The Niger catchment and maybe others have extensive wetlands which store and evaporate large amounts of water. Some of the catchments (e.g., the Colorado one) are also littered with reservoirs. All these factors contribute considerably to the total water storage and thus should be explicitly accounted for in such a study (as recognized by the authors at the end of the paper on page 17).

[Figure]

2. "We selected four basins as focus regions of study: the Colorado River basin in North America, Niger basin in Africa, Aral basin in Asia and the Lake Eyre basin in Australia (Figure 1)." Given the amount of hydrological data and computing resources available these days, why did you decide to focus on such a small sample of catchments?

3. Why are three products considered for evaporation and only one for precipitation? The differences among products are probably even larger for precipitation than for evaporation. Other viable precipitation products include WFDEI, MSWEP, and GPCC.

4. Why did you use simulated rather than observed runoff? NOAH is an uncalibrated model which means its simulated runoff contains biases.

5. The study concludes that there is little consistency among the different hydrological variables in the catchments. This is really not a novel result. Previous (global-scale!) papers have arrived at similar conclusions for evaporation (e.g., Miralles et al., 2016, HESS, cited in the paper) and precipitation (e.g., Herold et al., 2015, GRL). What new lessons we can learn from this paper?

6. The abstract is very poor. Only the last sentence can be considered results/discussion/conclusions. The previous sentences (with the exception of maybe the second to last sentence) are all introduction. It is not even mentioned how many catchments have been examined. Stick to one or maybe two sentences introduction.
* * *

---

## Author Comment (AC1) · 26 Jul 2016

As much as we would like to expand the analysis to include this interesting model, the work presented here focused on the period 2003-2011 to maximise the use of GRACE data. To our knowledge, the PT-JPL model does not extend to this period on a global basis. If there are more recent results, we would be open to include these in our analysis.

---

## Author Comment (AC2) · 26 Jul 2016

Author Introduction. Thank you for your positive comments and thoughtful review. In addressing these, we have itemized the various comments in order of their appearance in the letter.

Comment 1. This paper describes an evaluation method to explore correlation of satellite-derived water budget components P-ET and Terrestrial Water Storage. It uses a spherical harmonic analysis to analyse correlation and differences between multiple time series of P-ET and GRACE Terrestrial Water Storage. The method results cannot explain differences of the analyses in the three large-scale basin studies. This study uses a novel approach to estimate differences between two time series. The fact that this analyses does not lead (yet) to valuable results, is no reason to disqualify it in any

way. Therefore, it makes this paper a valid and potentially useful addition to the journal. However, the descriptions and approach are, in my opinion, not fully crystallised and require more research. Some of the descriptions are also rather confusing and need a better structuring (see further comments below). I get the feeling that recommendations described in this work should have actually be part of the paper.

Author response. The overriding objective of this work is to determine whether hydrological consistency can be achieved using independent remote sensing data over regions that exhibit "idealized" conditions (i.e. where hydrological processes are reasonably well defined). The fact that a high degree of consistency was not observed is an important result, as it highlights the challenge and disparity of these remote observations. The reason that we explored this approach was a means to independently assess a selection of evaporation models. Assessment of such large scale products is challenging, and new approaches are required to provide a more holistic evaluation strategy. This point may not have been well articulated, so we will re-examine the descriptions and rationale behind the approach to ensure greater clarity of purpose.

Comment 2. The whole term 'hydrological consistency', which is everywhere in the paper, e.g. the title, is not very clearly explained. Could you consider a better way to describe what you want to research? For example, 'the ability to balance the water budget' or something containing the words 'hydrological closure'?

Author response. This point is well taken. Remote sensing offers a number of independent means with which to retrieve various components of the hydrological cycle (i.e. rainfall, soil moisture, evaporation). Ideally, these observations should be hydrologically consistent: that is, an observed rainfall event should cause a corresponding change in soil moisture, for instance. Likewise, a reduction in soil moisture should be reflected by an increased flux of evaporation. Consistency is just another term that encompasses the expectation of a water budget: changes in one term should be reflected in others. While this has been explored qualitatively in the past (McCabe et al. 2008), here we wanted to determine if the method (using spherical harmonics) could reveal

some level of agreement between spatial (and temporal) patterns of these independent hydrological variables. We will ensure that this term is better defined so that the reader understands the intent (and limitations) of the research.

References

McCabe, M., Wood, E., Wójcik, R., Pan, M., Sheffield, J., Gao, H. and Su, H.: Hydrological consistency using multi-sensor remote sensing data for water and energy cycle studies, Remote Sensing of Environment, 112(2), 430-444, doi:http://dx.doi.org/10.1016/j.rse.2007.03.027, 2008.

Comment 3. In terms of concise descriptions: in my opinion, this study knows too many research questions and too little answers. It needs serious work on structure, correct descriptions and conciseness. The authors describe several paragraphs saying: 'the objective of this work', or 'a secondary objective of this work was..', or 'one key motivation of the study..'. Or they reword their aims in questions, such as: 'how accurate are the hydrological components derived from satellite observations?' or 'is hydrological consistency achieved with a particular product....?, etc. To me, that makes it confusing. Can you please clearly state the objectives and motivation at first, and then relate back to exactly those? That would make this work better readable. Furthermore, the many different aims and questions also makes you wonder about the conciseness and correctness of this study. For example, the questions "how accurate are the hydrological components derived from satellite observations?" cannot be answered with the results of this study. If all these research objectives and questions would be compiled into only two research questions, I think the paper would be more understandable.

Author response. We will work on restructuring the paper so that the main purpose and objectives can be identified more easily. As mentioned above, the goal is to evaluate the "hydrological consistency" method in a close-to-ideal scenario, keeping in mind that the desired potential use of the approach was to evaluate different evaporation products. How to evaluate large scale remote sensing products is an important ques-
tion that has some philosophical, as well as analytical, challenges. However, it may be better to ensure that these are contained in the discussion section rather than the introduction.

Comment 4. It is unclear what the added value of the spherical component analysis is. After all, correlations between water budget components can also be made in a different way, and the spherical component analyses does not lead to any new insights.

Author response. Undertaking the analysis in spherical harmonic space does seem like an unusual approach: but this is part of the novelty of the work. The reason for doing this was to ensure that a fair comparison between GRACE data and satellite products could be undertaken. Since GRACE data is filtered in spherical harmonics (unlike more traditional remote sensing variables in hydrology), a comparison between this and the other spatial maps of hydrological retrievals is likely to be imprecise (see Tapley et al., 2004 and supporting online material). Ensuring a consistency in spatial comparisons is one of the main reasons for doing the analysis in this way. While scaling the GRACE signal to account for differences has been proposed as an alternative solution to this problem (Landerer and Swenson, 2012), it has recently been shown to affect results (Long et al., 2015). By removing the impact (and model dependence) of this scaling term on the GRACE data, a more reasonable intercomparison of hydrological variables can be undertaken.

References

Tapley, B. D., Bettadpur, S., Ries, J. C., Thompson, P. F. and Watkins, M. M.: GRACE measurements of mass variability in the earth system, Science, 305(5683), 503-505, doi:10.1126/science.1099192, 2004. Landerer, F. W. and Swenson, S. C.: Accuracy of scaled GRACE terrestrial water storage estimates, Water Resources Research, 48(4), doi:10.1029/2011WR011453, 2012. Long, D., Longuevergne, L. and Scanlon, B. R.: Global analysis of approaches for deriving total water storage changes from GRACE satellites, Water Resources Research, doi:10.1002/2014WR016853, 2015.

Comment 5. Moreover, the lag between some data make the results of the analysis method less obvious. The method does not lead to any quantifications of if the water budget is in balance, or by how much it is off (for example, in percentages of the total water budget). That makes this study for me hard to judge: it evaluates water budgets [but] this does not lead to any new insights (unless you describe that part better), and cannot answer obvious explanations for imbalance of a water budget.

Author response. It is true that at this exploratory stage of investigation, the approach does not provide a quantified metric of water budget closure, since it is difficult to relate the correlation between the two sets of spherical harmonics into a measurable quantification of the water budget imbalance. However (and this relates a little to the philosophical aspect of product evaluation mentioned above), to attempt to do this by using observations alone (i.e. not involving a hydrological model) requires that the individual products are themselves well quantified (or "validated"). The reality is that at the large scales studied here (and even at much smaller scales), they are not. Large scale retrievals of evaporation, soil moisture and rainfall products do not come with well-defined accuracy metrics, let alone uncertainty bounds. The question of how to evaluate such datasets remains an outstanding one – and one that requires examining a range of approaches. Determining whether these individual products are at the least consistent with each other (i.e. they reflect hydrological expectation) is a needed first step in product assessment. That is essentially what we attempt to do here: and find (perhaps not surprisingly) that we are not able to do this yet, even in relatively simple systems. We believe that this is an important, if under-appreciated, insight that provokes a need for both improved products and evaluation strategies.

Comment 6. It is unclear why the focus is put on the ET component. You should pitch that better. After all, any uncertainty of the P component would result in larger uncertainty. I think the discussion and conclusion of this study need to point out that comparisons in catchment study need to be undertaken using the regional information on hydro(geo)logy and ground estimates of P, ET and streamflow. My guess is that you

want to say that, despite being the second-largest component of the water budget, ET is most uncertain?

Author response. The focus on the ET component comes from a desire to evaluate some recently developed global satellite evaporation products (see McCabe et al. 2016). Given the spatial mismatch between ground observations (and the lack of continuous large-scale coverage of in-situ data in remote regions), it is difficult (moreover, inappropriate) to validate these large-scale products in such a traditional manner: hence the comment on product evaluation being an outstanding problem in hydrology mentioned above. In the recent literature, inter-comparison between evaporation products has mostly been done by providing estimates of the uncertainty in terms of the variance among the products, sometimes including other types of datasets as well (from land surface models and climate reanalysis) [Jimenez et al., 2011; Mueller et al., 2011; Long et al., 2014]. While this is a good first order approach, it also recognises the challenge and lack of a benchmark evaluation set. Furthermore, rather than comparing the uncertainties between the evaporation products and the other hydrological components (which are poorly defined), we attempted to distinguish between the different evaporation products relative to their consistency with precipitation and storage. That is, are observed changes or patterns in the evaporation datasets reflected in these other hydrological variables. We explore this approach precisely because of the challenges in quantifying uncertainty based upon traditional in-situ methods.

In terms of exploring a range of other datasets, the GPCP product was chosen due to its global coverage as well as being widely used in the literature. However, as the reviewer notes, other precipitation products could be included to examine their impact. We have recently investigated this using the PERSIANN global data set (Hsu et al. 1997), but have not seen any significant change that would alter the conclusions of the study (see these early results and a preliminary Figure S1 below, that can be included as supplementary material). Further data sets can be considered if required, but ideally we would prefer that the focus remains on determining variability in the evaporation products rather than including a greater number of variables and complicating the analysis and interpretation of results.

References

Hsu, K.-l., Gao, X., Sorooshian, S. and Gupta, H. V.: Precipitation estimation from remotely sensed information using artificial neural networks, Journal of Applied Meteorology, 36(9), 1176-1190, 1997. Jimenez, C., Prigent, C., Mueller, B., Seneviratne, S. I., McCabe, M. F., Wood, E. F., Rossow, W. B., Balsamo, G., Betts, A. K., Dirmeyer, P. A., Fisher, J. B., Jung, M., Kanamitsu, M., Reichle, R. H., Reichstein, M., Rodell, M., Sheffield, J., Tu, K. and Wang, K.: Global intercomparison of 12 land surface heat flux estimates, Journal of Geophysical Research: Atmospheres, 116(D2), doi:10.1029/2010JD014545, 2011. Long, D., Longuevergne, L. and Scanlon, B. R.: Uncertainty in evapotranspiration from land surface modeling, remote sensing, and GRACE satellites, Water Resources Research, doi:10.1002/2013WR014581, 2014. McCabe, M. F., Ershadi, A., Jimenez, C., Miralles, D. G., Michel, D. and Wood, E. F.: The GEWEX landFlux project: Evaluation of model evaporation using tower-based and globally gridded forcing data, Geoscientific Model Development, 9(1), 283-305, doi:10.5194/gmd-9-283-2016, 2016. Mueller, B., Seneviratne, S. I., Jimenez, C., Corti, T., Hirschi, M., Balsamo, G., Ciais, P., Dirmeyer, P., Fisher, J. B., Guo, Z., Jung, M., Maignan, F., McCabe, M. F., Reichle, R., Reichstein, M., Rodell, M., Sheffield, J., Teuling, A. J., Wang, K., Wood, E. F. and Zhang, Y.: Evaluation of global observations-based evapotranspiration datasets and IPCC AR4 simulations, Geophysical Research Letters, 38(6), doi:10.1029/2010GL046230, 2011.

Comment 7. The word groundwater is mentioned in the description of GRACE data. However, it is surprising that the word groundwater is not mentioned when discussing the phase lag to GRACE data, nor the separation of the P-ET(=Q) into streamflow and groundwater flow. One could for example compare global data of baseflow (BFI) and look if these compare to the differences in lag. See for example the wonderful work of Beck et al. (2013): Beck, H. E., A. I. J. M. van Dijk, D. G. Miralles, R. A. M. de Jeu, L.

A. Bruijnzeel, T. R. McVicar, and J. Schellekens (2013), Global patterns in base flow index and reces- sion based on streamflow observations from 3394 catchments, Water Resour. Res., 49, 7843–7863, doi:10.1002/2013WR013918.

Author response. We agree that this is a very nice study and appreciate the reviewer drawing this paper to our attention. After examining the paper, it may prove useful in aiding the interpretation of the physical aspects of incorporating a lag term into this analysis i.e. comparing the BFI of the four regions of study (and streamflow parti- tioning), against the observed lag seen in our method to determine any similarity (or differences). Even though the BFI from this study is a static variable in time, we agree that it will likely add value to our discussion.

Comment 8. It is also surprising that nothing is said on snow storage in the discussion on P-ET.

Author response. Snowmelt resulting from snow storage in two of the study regions was recognized as one of the complicating elements in our study. Unfortunately, the type of basin that would yield the most useful results to our study is not, in practise, easy to find e.g. one that is large enough for GRACE to detect the change in mass, that has sufficient variability in water storage (a requirement for GRACE), while having a small or negligible runoff component, no snow component, and strong precipitation and evaporation fluxes! The most limiting factor in using GRACE data in the study is the size requirements for the studied catchment. Because of their large extent and ge- ographical features, the Colorado River and Aral Sea basins do include regions where snow storage plays an important role. Snow storage itself is not a problem, because GRACE detects changes in storage irrespective of their nature (snow, groundwater, soil moisture, etc). However, snowmelt may contribute to delayed changes in storage that can affect the results. The influence that snowmelt, as well as other potential sources of lag in the system is not known, and forms part of reason to explore the inclusion of a lag in the GRACE data.

Comment 9. Furthermore, the description of input data is not clear. Especially the ET part should contain what part of the data we use. For example, do we use PET from MOD16 or AET? Also, MOD16 contains an unclear description that first suggests it is a Penman- Monteith method, and then suggests it is Priestley-Taylor. Furthermore, the spatial detail in Table 1 on MOD16 is not correct to my knowledge and resolution should be 1km (at least if the data from Mu et al from http://www.ntsg.umt.edu/project/mod16 is used). See more detailed text comments below.

Author response. In this work, actual evapotranspiration (AET) was used for all evaporation products: this should have been more clearly articulated in the text. The particular sentence the reviewer refers to (Page 5, line 10) will be rephrased in a revised version to clearly indicate that Priestley-Taylor is only used for plant transpiration. In terms of the resolution, MOD16 (as found in http://www.ntsg.umt.edu/project/mod16) is available as a 1 km resolution product in a sinusoidal projection. In this study, the 1 km sinusoidal product was reprojected to a regular 0.05 degree-resolution grid using the MODIS Reprojection Tool (MRT). While a 0.05 degree resolution monthly product was available, we summed the original product to get monthly estimates in order to be time-consistent with the GRACE CSR product months.

We appreciate the potential confusion, so will improve the explanation in a revised version.

Comment 10. Methods and results are in my opinion too intertwined, and should be separated more. For example, in the results, there are some descriptions of the basins. For example, descriptions of the CRB, ASB and LEB basins contain texts that should in my opinion be put into 2.5 Study Regions.

Author response. We will review the manuscript and where needed, better separate background methods from results. Some of the description of the basins found in the results relate to the observed trends in water storage and precipitation observed during the study period. This was done in order to relate the changes in degree correlation to

the changes during these trends.

Comment 11. I am not an expert in spherical harmonic analyses, so I cannot judge on the correctness of the method. However, I think it would make the text a bit clearer if the term 'degree' (l) would be explained a bit more in detail, so non-experts in spherical harmonics would also understand. In that way, your plots 4-7 would be easier to understand.

Author response. We acknowledge that this is an abstract concept and we will work on the phrasing in the methodology to better explain it to the broad readership of HESS.

Comment 12. A weakness of this paper is that it does not incorporate uncertainty. After all, estimates of ET using ground observations already contain quite a bit of uncertainty (see e.g. Westerhoff, 2015, with work on Uncertainty of Penman and Penman-Monteith estimates). Another weakness of this paper is that it is not clear what specific ET has been used (PET, AET, P-T, P-M). A quick fix could be a better description of the input data. However, a discussion of different methods used (e.g. P-T or P-M, or PET or AET etc) should be clear on pointing out differences in methods. It would be unfair to judge that satellite data is not hydrologically consistent, if the ground-estimates already are so uncertain and if the different methods have caused this uncertainty.

Author response. As mentioned above, actual evaporation from all three evaporation products is used. We fully appreciate the comment regarding the uncertainty of both satellite and ground based measurements of evaporation. Indeed, our group have published a number of papers on this precise topic. Given the already extensive literature on this, we are not convinced that further repetition (or additional model descriptions) will add much to the paper. We will certainly attempt to clarify the precise nature of the products used by ensuring a thorough review of the input descriptions and can add detail to data and model descriptions where required.

The quality issue of ground based observations that you correctly raise is one reason why alternative approaches to product evaluation are required. The quality (or

otherwise) of the satellite product should not be judged (solely) on its agreement with unrepresentative point scale approaches. It should also be judged on whether it can reflect hydrological expectation, as observed in independently observed and hydrologically linked variables (e.g. rainfall, soil moisture or groundwater storage). It is this approach that we have adopted to explore here. Certainly a more holistic evaluation strategy is required to ensure greater confidence in large scale products.

Comment 13. It is also unfair to put 'the blame' on ET, because P is a much more likely candidate for this uncertainty. You need to describe better why ET is so important in this analyses. My guess is that you want to say that, despite being the second-largest component of the water budget, ET is most uncertain. Can you derive uncertainties of all estimates? This study would be much stronger if it could quantify what absolute values of inconsistencies we are talking about.

Author response. As mentioned earlier, the focus on evaporation in this study was based on a need for an alternative way to evaluate satellite evaporation products (since in-situ observations are not the best way forward). Therefore, only a single precipitation product was explored: albeit one that has been well studied and used in global analysis. But this is also true for the water storage variations: although there are a number of water storage products based on GRACE satellite data (Bruinsma et al., 2010; Liu et al., 2010; Rowlands et al. 2005), the choice of a particular product (CSR) was based on its widespread use in a number of previous studies. Having said this, other precipitation products can certainly be considered and can be explored in a revised version (see earlier comment and preliminary results with PERSIANN).

References

Bruinsma, S., Lemoine, J.-M., Biancale, R. and Valès, N.: CNES/GRGS 10-day gravity field models (release 2) and their evaluation, Advances in Space Research, 45(4), 587-601, doi:http://dx.doi.org/10.1016/j.asr.2009.10.012, 2010. Liu, X., Ditmar, P., Siemes, C., Slobbe, D. C., Revtova, E., Klees, R., Riva, R. and Zhao, Q.: DEOS mass transport

model (DMT-1) based on GRACE satellite data: Methodology and validation, Geophysical Journal International, 181(2), 769-788, doi:10.1111/j.1365-246X.2010.04533.x, 2010. Rowlands, D. D., Luthcke, S. B., Klosko, S. M., Lemoine, F. G. R., Chinn, D. S., McCarthy, J. J., Cox, C. M. and Anderson, O. B.: Resolving mass flux at high spatial and temporal resolution using GRACE intersatellite measurements, Geophysical Research Letters, 32(4), doi:10.1029/2004GL021908, 2005.

Comment 14. If the document would be structured better, I think it would improve the quality and probably I would understand better what the exact goal is of this paper. Reducing the number of research questions/objectives would probably clarify why this paper should be published. Therefore, I recommend a major revision.

Author response. We appreciate your thoughtful and constructive comments. We will make sure to structure the text throughout the document so that the main goal of the study can be understood clearly, as well as better articulating the results and novelty of this work.

Figure S1. Top: average degree correlation statistics per study region and evaporation product. Bottom: GRACE data were shifted by two months to match the phase with P-E anomalies. The boxplots show the first, second (median) and third quartiles. Outliers, defined as data outside the 1.5 inter-quartile range (IQR) whiskers below or above the first and third quartiles are shown as circles. This figure represents a summary of the analysis using the PERSIANN product as precipitation. **The results are very similar to those in Figure 8.**

**Fig. 1.** Supplementary Figure 1

---

## Author Comment (AC3) · 26 Jul 2016

Author Introduction. We thank the anonymous reviewer for their insightful comments on our manuscript. We have attempted to address these points in the responses below.

Comment 1. The study focuses on four "relatively simple" catchments. But what's so simple about these catchments? The runoff fraction is definitely not negligible (see Figure 2). The snowfall fraction in some of them is quite high (e.g., precipitation in the Aral Sea and Colorado catchments is composed of about 15% snowfall). The Niger catchment and maybe others have extensive wetlands which store and evaporate large amounts of water. Some of the catchments (e.g., the Colorado one) are also littered with reservoirs. All these factors contribute considerably to the total water storage and thus should be explicitly accounted for in such a study (as recognized by the authors

at the end of the paper on page 17).

Author response. We agree that these are not "ideal" catchments to explore our approach. Unfortunately, the requirements for a candidate basin that would yield the most useful results to the study made the selection of basins difficult: a basin that is large enough so that its mass changes can be detected by GRACE satellites, yet with a lack of complicating factors contributing to the hydrological balance (see additional response to another reviewer). Ultimately, we compromised to "relatively simple", since in comparison to many other basins, the hydrological processes in arid and semi-arid environments are somewhat better defined. In terms of water storage (either in reservoirs, wetlands or as snow), this should not be a major issue as GRACE is able to detect large changes in mass irrespective of their source. Of course, in this case the problem lies when the mass loss is not accounted for by evaporation. This was recognized in the text as one of the challenges in both the Aral Sea and Colorado River basins and remains a problem in large scale evaporation retrievals.

Comment 2. "We selected four basins as focus regions of study: the Colorado River basin in North America, Niger basin in Africa, Aral basin in Asia and the Lake Eyre basin in Australia (Figure 1)." Given the amount of hydrological data and computing resources available these days, why did you decide to focus on such a small sample of catchments?

Author response. As noted above, the major limitation was the size requirement due to the coarse resolution of GRACE data. This need, coupled with the limitation of a "relatively" simple hydrological system, limited the options for basin selection. However, even an analysis focusing on these four basins provides a good initial framework for assessing the utility of the approach. While the approach could be extended to include all candidate basins globally, our preference was to focus on a selected number that we could reasonably characterise to explore the potential. We would note that there are numerous examples in the literature of analysis of GRACE products and other data being examined over just a single basin. Future work can certainly expand the

technique to other basins and climate types.

Comment 3. Why are three products considered for evaporation and only one for precipitation? The differences among products are probably even larger for precipitation than for evaporation. Other viable precipitation products include WFDEI, MSWEP, and GPCC.

Author response. The choice for a single precipitation product (as well as only one GRACE product) was intentional, as the study focused on a desire to explore the quality of a number of competing satellite evaporation products. The method was originally targeted as an evaluation tool for these large-scale retrievals. We sought to determine whether differences in evaporation, while constraining the variability in other hydrological components, could be detected by the method. The GRACE CSR product was chosen as it is the most commonly used of the three official products. In terms of precipitation, GPCP was chosen due to its global coverage, temporal extent and extensive use in the literature. However, we can certainly expand the analysis to include other precipitation products. So far, we have seen that the conclusions of the study are not altered when using the PERSIANN dataset (Hsu et al., 1997) (see preliminary results in Figure 1 below for inclusion as supplementary material).

References

Hsu, K.-l., Gao, X., Sorooshian, S. and Gupta, H. V.: Precipitation estimation from remotely sensed information using artificial neural networks, Journal of Applied Meteorology, 36(9), 1176-1190, 1997.

Comment 4. Why did you use simulated rather than observed runoff? NOAH is an uncalibrated model which means its simulated runoff contains biases.

Author response. This seems to be a misunderstanding. Neither simulated nor observed runoff were explicitly included in the evaluation of hydrological consistency. Simulated runoff from GLDAS-NOAH was plotted in Figure 2 only to review the assumption of a 'relatively' simple water cycle in arid and semi-arid regions. As was noted in addressing the first comment, this was not always the case for the chosen study basins and the implications of this were described in the Discussion section. We will ensure that this point is more clearly articulated in the relevant text.

Comment 5. The study concludes that there is little consistency among the different hydrological variables in the catchments. This is really not a novel result. Previous (global-scale!) papers have arrived at similar conclusions for evaporation (e.g., Miralles et al., 2016, HESS, cited in the paper) and precipitation (e.g., Herold et al., 2015, GRL). What new lessons we can learn from this paper?

Author response. While a number of global evaporation (and precipitation) evaluation papers have recently been published (including McCabe et al. 2016), they do not seek to compare consistency with related hydrological variables, but instead focus largely on an assessment against point-scale observations and/or inter-product variability. Our approach is designed to see whether within these varying products there is one (or more) that are at least consistent with other related hydrological variables (in this case, P and dS). This is a major distinction to past work. Importantly, many of these recent evaporation assessment papers explicitly identify the problems of employing a tradi-tional point-scale evaluation approach and the need to explore alternative assessment techniques (such as that developed here).

As such, we propose a quite different method for assessment based on the concept of hydrological consistency. As referenced in the introduction (and elsewhere in the paper), a limited number of water budget studies, ranging from individual basins (Pan et al., 2008; Sheffield et al. 2009) to global scale studies (Sahoo et al., 2011, Pan et al., 2012), have shown large errors in water budget closure: but have done so with a focus entirely on the temporal scale and invoking the use of a hydrological model. Some of these studies (Sahoo et al, 2011; Pan et al., 2012) sought to constrain the water balance by using a merged product or data assimilation approach, but did not evaluate hydrological consistency of the individual products. This is what is explored

in our work. The motivation behind this study was to take a step back and determine whether a first order hydrological agreement could be achieved in these relatively simple environments. As was found, the challenge on how to do this remains, raising some important questions on both product quality and also the techniques we use to evaluate global products. We will attempt to better articulate these issues and other novel aspects.

References

McCabe, M. F., Ershadi, A., Jimenez, C., Miralles, D. G., Michel, D. and Wood, E. F.: The gEWEX landFlux project: Evaluation of model evaporation using tower-based and globally gridded forcing data, Geoscientific Model Development, 9(1), 283-305, doi:10.5194/gmd-9-283-2016, 2016. Pan, M., Wood, E. F., Wójcik, R. and McCabe, M. F.: Estimation of regional terrestrial water cycle using multi-sensor remote sensing observations and data assimilation, Remote Sensing of Environment, 112(4), 1282-1294, doi:http://dx.doi.org/10.1016/j.rse.2007.02.039, 2008. Pan, M., Sahoo, A. K., Troy, T. J., Vinukollu, R. K., Sheffield, J. and Wood, E. F.: Multisource estimation of long-term terrestrial water budget for major global river basins, Journal of Climate, 25(9), 3191-3206, doi:10.1175/JCLI-D-11-00300.1, 2012. Sahoo, A. K., Pan, M., Troy, T. J., Vinukollu, R. K., Sheffield, J. and Wood, E. F.: Reconciling the global terrestrial water budget using satellite remote sensing, Remote Sensing of Environment, 115(8), 1850-1865, doi:http://dx.doi.org/10.1016/j.rse.2011.03.009, 2011. Sheffield, J., Ferguson, C. R., Troy, T. J., Wood, E. F. and McCabe, M. F.: Closing the terrestrial water budget from satellite remote sensing, Geophysical Research Letters, 36(7), doi:10.1029/2009GL037338, 2009.

Comment 6. The abstract is very poor. Only the last sentence can be considered results/discussion/conclusions. The previous sentences (with the exception of maybe the second to last sentence) are all introduction. It is not even mentioned how many catchments have been examined. Stick to one or maybe two sentences introduction.

Author response. We will review the abstract, and if necessary, iterate to present a clearer description of the work and key results.

[Figure]

[Figure]

Figure S1. Top: average degree correlation statistics per study region and evaporation product. Bottom: GRACE data were shifted by two months to match the phase with P-E anomalies. The boxplots show the first, second (median) and third quartiles. Outliers, defined as data outside the 1.5 inter-quartile range (IQR) whiskers below or above the first and third quartiles are shown as circles. This figure represents a summary of the analysis using the PERSIANN product as precipitation. **The results are very similar to those in Figure 8.**

**Fig. 1.** Supplementary Figure 1

---

## Author Response (AR1)

**Comments to the authors by Prof. Dr. Marc Bierkens**

This paper seeks to evaluate the consistency between independently obtained remotely sensed water cycle products and at a comparable spatio-temporal scale. However, the focus is on Evap as 3 Evap products are used and one GRACE and one Precip product. The novelty of this study is to look at consistency instead of uncertainty and/or water balance closure. This by itself is worthy for publication. However, there are a number of issues that need to be resolved as brought forward by the two reviewers. The most important issues are with the structure and readibility of the paper which could be much improved. Most significantly:

1) The term "hydrological consistency" should be better defined.

2) The authors should more clearly state the objective(s) of the paper, state it only in one paragraph and refer to that later in the discussion of the results.

In their rebuttal the authors already provide a much clearer description of their purpose:

"Ideally, these observations should be hydro- logically consistent: that is, an observed rainfall event should cause a corresponding change in soil moisture, for instance. Likewise, a reduction in soil moisture should be reflected by an increased flux of evaporation. Consistency is just another term that en- compasses the expectation of a water budget: changes in one term should be reflected in others. While this has been explored qualitatively in the past (McCabe et al. 2008), here we wanted to determine if the method (using spherical harmonics) could reveal some level of agreement between spatial (and temporal) patterns of these independent hydrological variables."

After that they could then state the objective as" So the objective is to check this hydrologic consistency for a number of basins where hydrological processes are relatively well defined."

3) State clearly in the introduction what the novel features are of this work (checking consistency instead of water balance closure and/or uncertainty and doing this at a comparable spatio-temporal scale).

4) More clearly define what you mean by a "simple hydrological system" or "relatively well defined hydrological processes". What conditions have been used to select the basins.

5) Consider changing the title somewhat. As the focus is on Evap it could for instance be:

Evaluating the hydrological consistency among satellite based water cycle components with a focus on evaporation products

**Author's response to the Editor's comments**

Dear Prof. Dr. Marc Bierkens,

Thank you for your positive comments. In revising our manuscript, we have addressed all of the thoughtful and insightful comments put forward by the reviewers, as well as responded to the five major points you have detailed in your correspondence.

Following your suggestion, we more clearly state that the focus of the paper is to evaluate evaporation products. To reflect this, the title of the paper has been changed to: "Evaluating the hydrological consistency of evaporation products using satellite-based gravity and rainfall data"

Changes have also been reflected in a revised Abstract, Introduction, and Discussion sections, which have been subject to considerable and focused modification. Some of these are summarized below, followed directly by a complete list of changes.

- In the Introduction Section, the manuscript now addresses the challenges and issues associated with assessing large-scale evaporation products, and we have included further details as a result of the discussion with the reviewers. The definition of hydrological consistency is now more clearly articulated. The objective of the study, which is related to employing this concept of hydrological consistency, is restated in the revised Introduction.

- Another issue that was highlighted in your response was to clearly state the conditions that have been used to select basins with relatively simple interactions. We have made sure to include these early in the manuscript. In addition, we have more clearly highlighted the novelties of the study, which include (in part) the approach of hydrological consistency to evaluate evaporation products, as well as the spherical harmonics degree correlation approach to intercompare the products.

- Related to this last point, we have responded to the reviewer's request for a more detailed explanation of the spherical harmonics concept. These include:

  - From the Methodology Section, Lines 20 in page 7 through line 11, page 8, which introduce some of the theory pertaining to GRACE and the use of spherical harmonics, have been moved up to the Data Section under the subsection "GRACE water storage anomalies".

  - The introduction now contains a brief, but necessary introduction to this concept (to further highlight the novelty aspects of the paper).

  - In addition, lines 33 (page 2) through line 9 (page 3) have been moved to the Data section under the subsection "GRACE water storage anomalies"

- A description of the evaporation (and precipitation) data sets was also revised and now includes information on the inputs requirements, as well as a more detailed description. In this section, we have also made sure to better explain the selection process for the study basins. Some additional details on the selected basins that were previously included in the Results were moved to the Data section.

- In the Results Section, we have added some comments on the interpretation of the lag phase found in GRACE data. This includes new information pertaining to the base flow index (BFI; Beck et al., 2013), as suggested by one of the reviewers.

- The Discussion Section was also extensively revised. We have now divided this Section under three sub-headings: (1) Challenges to implementing hydrological consistency, (2) Temporal lag in terrestrial storage response, and (3) Discriminating between satellite evaporation products. We believe this will allow the reader to more clearly identify and appreciate the discussion generated from our study. We have also added a supplementary Figure and reference to it in the Discussion Section in order to address the use of another precipitation product in the study, as suggested by both reviewers.

- Finally, the abstract was also extensively revised based on the comments by the reviewers. The abstract has now a briefer introduction, and focuses more on the novelty and results of the study.

We believe that this newly revised manuscript better presents the findings of our study. We thank you for time and consideration.

**List of changes to the manuscript**

**Abstract**

Page 1, lines 7-9.  Modified text:

Advances in space--based observations  have provided the capacity to develop regional- -to -global--scale estimates of evaporation,  offering insights into  this key component of the hydrological cycle.

Page 1, lines 9-22 Removed completely, inserted:

> However, the evaluation of large-scale evaporation products is not a straightforward task. While a number of studies have intercompared a range of these products by examining the variance amongst them or by comparison of pixel-scale retrievals against ground-based observations, there is a need to explore more appropriate techniques to comprehensively evaluate flux estimates. One possible approach is to establish the level of product agreement between related hydrological components: for instance, how well do evaporation patterns and response match with precipitation or water storage changes. To assess the suitability of this "consistency"-based approach for evaluating evaporation products, we identified four globally distributed basins in arid- and semi-arid environments, including the Colorado River basin, the Niger River basin, the Aral Sea basin and the Lake Eyre basin. In an effort to identify product quality, three satellite-based global evaporation products including CSIRO-PML, MOD16 and GLEAM were evaluated against rainfall data from GPCP along with GRACE water storage anomalies. To ensure a fair comparison, we evaluated consistency using a degree correlation approach after transforming both evaporation and precipitation data into spherical harmonics.

Page 1, lines 22-24

Overall we found~~, it makes sense to first test it over environments with restricted hydrological inputs, before applying it to more hydrological complex basins. Here we explore the concept of hydrological consistency, i.e. the physical considerations that the water budget impose on the hydrologic fluxes and states to be temporally and spatially linked, to evaluate the reproduction of a set of large-scale evaporation (E) products by using a combination of satellite rainfall (P) and Gravity Recovery and Climate Experiment (GRACE) observations of storage change, focusing on arid and semi-arid environments, where the hydrological flows can be more realistically described. Our results indicate~~ no persistent hydrological consistency in these dryland environments.

Page 1, lines 24-25 Inserted:

> Indeed, the degree correlation showed oscillating values between periods of low and high water storage changes, with a phase difference of about 2-3 months. Interestingly, after imposing a simple lag in GRACE data to account for delayed surface runoff or baseflow components, there was an improved match in terms of degree correlation in the Niger River basin. Significant improvements to the degree correlations (from ~0 to about 0.6) were also found in the Colorado River basin for both the CSIRO-PML and GLEAM products (MOD16 showed only half of that

improvement). In other basins, the variability in the temporal pattern of degree correlations was still considerable and hindered any differentiation between the evaporation products. Even so, it was found that a constant lag of two months provided a better fit compared to other alternatives, including a zero lag. Regardless of this finding, from a product assessment perspective, it was concluded that no significant or persistent advantage could be identified across any of the three evaporation products in terms of a sustained hydrological consistency with precipitation and water storage anomaly data. The results of this analysis have implications in terms of the confidence that can be placed in independent retrievals of the hydrological cycle, raises questions on inter-product quality and highlights the need for additional techniques to evaluate large-scale products.

**Introduction**

Page 1, line 27 Inserted:

Space-based observations of the Earth system have provided the capacity to retrieve information across a wide-range of land surface hydrological components and an opportunity to characterize terrestrial processes in space and time. Indeed, remote sensing offers a number of independent means with which to retrieve various components of the hydrological cycle (e.g. rainfall, soil moisture, evaporation, terrestrial storage).

Page 1, line 27 Modified "observations" to "observation"

Page 2, line 3 regional- -to -global--scale

Page 2, line 4 Removed "key"

Page 2, line 5 Modified "is to characterize" to "is how to characterize"

Page 2, lines 6-8 Moved the following text to second paragraph in revised Introduction

Inherent to this challenge is the issue of scale, a consequence of both a lack of abundant high-quality in-situ data and the fact that there is an inevitable scale mismatch between these measurements (McCabe et al. 2006).

Page 2, lines 10-19 Modified in its entirety to the text below, and moved two paragraphs below (the following text constitutes the entirety of the fourth paragraph in the revised Introduction):

Observation-only studies are important, as they provide an unbiased perspective not just on hydrological closure, but also allow for a first-order examination of the underlying agreement between component variables. However, rather than just comparing the uncertainties between evaporation products and other hydrological components (which are poorly defined), there is still a need for alternative assessment techniques that explore the connection between the hydrological variables at both temporal and spatial scales. One approach to determine this is to evaluate the hydrological consistency between observed products (McCabe et al., 2008). The term hydrological consistency refers to the spatial and temporal match that should inherently exist between independent observations of hydrological states and fluxes, based upon physical considerations. It is a concept that encompasses the expectation of water cycle behavior and mass balance: that is, changes in one term should be reflected in related variables, both spatially and temporally. For instance, a rainfall event should result in an observable change in soil water storage and a

consequent increase in evaporative flux, which in turn should reduce the available soil moisture. This relatively simple concept has been explored in the past, including in efforts to improve precipitation events by employing cloud detection methodologies (Milewski et al., 2009); using soil moisture changes to infer precipitation amounts (Brocca et al., 2014); examining the connection between soil moisture state and changes in atmospheric variables such as humidity and sensible heat flux (McCabe et al., 2008); as well as in assessments of land–atmosphere coupling between observations and reanalysis data (Ferguson and Wood, 2010).

Page 2, lines 19-24 Modified and moved to next paragraph (this new text constitutes the entirety of the third paragraph in the revised Introduction):

> A crucial task that is required to address these questions is to evaluate the hydrological consistency amongst these different hydrological products. Hydrological consistency refers to the spatial and temporal match that must exist between individual observations of hydrological states and fluxes based on physical considerations. For example, cloud detection can be used to validate precipitation events (Milewski et al., 2009); soil moisture changes should closely match precipitation anomalies; changes in atmospheric related variables such as humidity and sensible heat flux should correspond to the soil moisture state (McCabe et al., 2008); and, in a larger scale, the spatial distribution and timing of water storage anomalies should be closely related to precipitation anomalies. In principle, in regions where runoff is low, independent estimates of water storage, precipitation and evaporation should provide a physically based closure of the water budget. However, even excluding the uncertainties inherent in the modeling and retrieval of these variables from satellite data, the complexities of land surface dynamics, conditions and residence times also make it challenging to apply in practice.Asoughtthes closure of large basins across different regions of the worldviathe objectiveof,was tothestreamflow values),the remainingmostly, and anyis~~given in terms of the variability among the products (e.g. Long et al., 2014). The results of these studies have generally illustrated large water budget closure errors, focusing on the temporal scale and invoking the use of a hydrological model to guide analysis or force closure, rather than being solely observation driven assessments.

Page 2, lines 24-31 Moved and modified (including additions to the text) the following text to fifth paragraph (this new text constitutes the entirety of the new fifth paragraph) in the revised Introduction.

[revised manuscript text omitted]

Page 2, line 33 – Page 3, line 4 Moved to beginning of subsection 2.1 (GRACE water storage anomalies)

Page 3, lines 6 – 19 Modified (see following changes below) and moved in its entirety to second paragraph in revised Introduction

Page 3, line 6 Modified "serves as a key component in our analysis." to:

"plays a key role in the water cycle as a linking mechanism between the surface and the atmosphere (Mueller et al., 2011)"

Page 3, line 7 Modified "or atmosphere that can be used" to "or atmosphere, which can be used"

Page 3, line 10 Modified "global scale" to "global-scale"

Page 3, lines 12 – 19 Modified text, including addition of moved text from Page 2, lines 6-8:

When ground-based flux observations are available, they can be used for calibration and evaluation, but large-scale assessment is inevitably constrained by the lack of distributed and representative in situ networks to comprehensively assess simulations (Jana et al., 2016)  as well as the inherent uncertainty associated with these observations. Some recent evaluation efforts have sought to estimate the uncertainty of satellite-based models of evaporation, as well as those from land surface model and reanalysis data, in terms of the variance amongst the products (Mueller et al., 2011; Jimenez et al., 2011; Long et al., 2014). These and related attempts have shown that no single evaporation product consistently outperforms any other, whether applied at local (Ershadi et al., 2014) or global scales (Miralles et al., 2016). Considering this issue of spatial mismatch and model variability, it seems inappropriate to assess these large-scale products via direct comparison to in situ data alone. Moreover, the quality of any satellite-based product should not be judged solely on its agreement with potentially unrepresentative point-scale approaches. Central to this challenge is the issue of scale, a consequence of both a lack of abundant high-quality in situ data and the fact that there is an inevitable scale mismatch between ground- and satellite--based observations (McCabe et al. 2006). To compensate for this, it is important that a range of methods be used to evaluate the large-scale implementation of evaporation models.

Page 3, lines 6-19 End of changes to this paragraph, which has been moved in its entirety to second paragraph in revised Introduction.

Page 3, lines 21 – 32 Modified in its entirety to:

The Overall objective of this study is to evaluate the hydrological consistency of independent satellite-based evaporation products with remotely sensed retrievals of precipitation and terrestrial water storage across a selection of basins that exhibit relatively well defined hydrological interactions. Throughout this analysis we aim to determine

whether the hydrological consistency concept can expand the range of evaluation metrics used to assess large-scale hydrological data sets such as evaporation, and enable some differentiation of relative product quality  to be made. ~~observations over basins where it is assumed that the water cycle system is relatively simple. Through this analysis we expect to determine whether the concept of hydrological consistency can be employed in regions with more complicated water cycle systems to aid in the validation of evaporation and other hydrological data sets. A secondary objective is to determine the impact that the choice of different evaporation products has on the analysis: is hydrological consistency achieved with a particular product, or is the disagreement between evaporation products significant enough to impact the study? Furthermore, if the hydrological consistency approach is not achievable, what does this say about the retrieval accuracy of these independent observations of the hydrological cycle? Section 2 describes the data sources, including a brief description of the global evaporation products used. Section 3 presents in detail the methodology used to evaluate the hydrological consistency based on a spherical harmonic analysis. The results in Section 4 show the spatial and temporal behavior of the correlations between water storage anomalies and P-E, while the implications of these are discussed in greater detail in Section 5. Finally, concluding remarks are provided in Section 6.~~

**Data and Methodology**

Page 4, line 4 Modified "evaporation products" to "recently developed global-scale evaporation products"

Page 4, line 8 Inserted text moved from Page 2, line 33 – Page 3, line 4

Page 4, line 8 Modified "derived from " to "computed using"

Page 4, line 9 Inserted the following text:

The gravity field is usually described in terms of the geoid: an equipotential surface that is defined to correspond to the mean sea level over the oceans (Swenson and Wahr, 2002). The geoid is usually approximated as a linear combination of spherical harmonics, given that these represent solutions to the Laplace equation that describes the relation between the gravitational potential and the geoid. The approximation is of the form:

Page 4, line 10 Inserted equation moved from Page 7, line 18 (i.e. equation 1)

Page 4, line 11 Inserted and modified text from Page 7, lines 19-22:

where $\tilde{P}_{lm}(\cos\theta)$ are the normalized associated Legendre functions, $\theta$ corresponds to the colatitude (the complementary angle to the latitude), $\phi$ to longitude, $C_{lm}$ and $S_{lm}$ are the spherical harmonic coefficients of degree l and order m,   and $l_{max}$ is the truncation degree. The total number of coefficients is given by $((l_{max} +1)^2 + l_{max}+1)/2$, while the resolution (the scale of the smallest feature of the gravity field that can be resolved using $l_{max}$ coefficients) is approximately $\pi a/l_{max}$ (where a is the Earth's radius). The data product used in this study (processed by UTCSR) contains coefficients up to $l_{max} = 60$, i.e. a total number of 1891 coefficients, with an approximate resolution of 333 km. The full description of the process to transform the gravity field anomalies into water storage anomalies is described  in Wahr (1998) and Swenson and Wahr (2002).

Page 4, lines 9-20 Modified text and moved to next paragraph (following text constitutes the new paragraph in its entirety):

GRACE data contains two types of errors (correlated and random) that need to be filtered before translating the data into water storage anomalies. Correlated errors are known to contaminate the signal in the form of north-south oriented stripes. A "de-striping" filter was applied to the coefficients (Swenson and Wahr, 2006; Duan et al, 2009) in order to remove this source of error. An isotropic filter (Gaussian filter with radius of 300 km) was then used to remove random errors (Swenson and Wahr, 2002). Furthermore, it is a usual practice to replace the  degree 2 coefficients with a more reliable estimate from a low-degree model of the gravity field calculated using  satellite laser ranging  (Cheng et al., 2011 Cheng et al., 2013).  While the effect that the filters has on the true geophysical signal is not known a priori, an indirect measure can be obtained by applying the filter to a synthetic water storage variation from a land surface model (LSM). This method has been used to obtain scaling factors for GRACE data in order to restore the signal (it has been observed that the filters typically reduce the signal) before using the GRACE data with other hydrological variables (Landerer and Swenson, 2012). Long et al. (2015) evaluated the impact of different land surface models on the scaling factor and showed that the impact was greatest in arid regions. To avoid this potential element of uncertainty in our study, which is focused on arid regions, we instead transformed the other water cycle components (i.e. evaporation and precipitation) into spherical harmonics, using an approximation similar to equation 1. The effect of the filters is therefore incorporated directly into the other hydrological components in spherical harmonics.

Page 4, line 26 Modified "described briefly" to "briefly described"

Page 5, line 3 Modified "Leaf Area Index" to "leaf are index"

Page 5, line 4 Removed "and Normalized Difference Vegetation Index (NDVI),"

Page 5, line 5 Added ", a three--source scheme used for terrestrial land flux estimation." after "(MOD16)"

Page 5, line 7-8 Modified "Other improvements include:" to "Other adjustments incorporated into MOD16 include"

Page 5, lines 9-10 Modified "The product includes transpiration and evaporation from soil and wet canopy," to "The MOD16 product comprises transpiration evaporation from the soil and wet canopy,"

Page 5, lines 10-11 Removed "Potential evaporation (calculated using a Priestley-Taylor based formulation) is also included to monitor environmental water stresses and droughts (Mu et al., 2011)"

Page 5, line 11 Added new text:

Each component is weighted-based on the fractional vegetation cover, relative surface wetness and available energy. Inputs to the model include net radiation (Rn), air temperature and humidity, as well as LAI and vegetation phenology. Importantly, it does not require wind speed or soil moisture data, making it a relatively parsimonious model in terms of input requirements. In this study, we used the actual evaporation (AET) product from MOD16 (Mu et al., 2011) with 8-day temporal resolution and 1 km resolution in the sinusoidal projection. The product was reprojected onto a 0.05° regular grid using the MODIS Reprojection Tool (MRT) before transformation into

spherical harmonics, as described in Section 3.1. Further details on the modeling basis behind the MOD16 product can be found in Mu et al. (2013), Ershadi et al. (2014) and Michel et al. (2016).

Page 5, line 14 Modified "resulting in the Penman-Monteith-Leuning (PML) model" to "resulting in the two-source Penman–-Monteith–-Leuning (PML) model"

Page 5, line 15 Modified "canopy conductance." to "the surface resistance, which was previously calculated as LAI multiplied by a constant $c_L$ (Cleugh et al., 2007)."

Page 5, lines 15-19 Removed completely and replaced by the following new addition:

The new parameterization of the surface resistance in the PML model was optimized using data from 15 globally distributed flux station sites, with two key parameters identified: the maximum stomatal conductance (gsx) and the ratio of actual to potential evaporation at the soil surface. Zhang et al. (2010) developed a method to further optimize the spatial variability of these two parameters (i.e. at each grid pixel) using gridded meteorological data and a simple Budyko-curve hydrometeorological model developed by Fu (1981) that includes precipitation and available energy as inputs. Mean annual evaporation for each grid pixel is calculated using the Fu model and gridded meteorological data. The value of gsx is optimized using a non-linear least square regression-based on the difference between the PML and the Fu model. Interestingly, the Fu model is calibrated by comparing the output evaporation with the residual of precipitation and runoff i.e. by assuming negligible annual water storage changes and groundwater inflow and outflow. Zhang et al. (2012) used this approach to develop a global gridded terrestrial evaporation product (hereafter referred to as CSIRO-PML; Zhang, 2014, personal communication) with a 0.25° resolution (in this study, we used the actual evaporation product). They used gridded meteorological data from diverse sources, including vapor pressure and temperature from the Climate Research Unit (New et al., 2000), LAI and land cover type from Boston University (Ganguly et al., 2008), precipitation from the Global Precipitation Climatology Centre (GPCC, version 4; Rudolf and Schneider 2004), and radiation data from the Global Energy and Water Cycle Exchanges (GEWEX) Surface Radiation Budget (Gupta et al., 2006).

Page 5, line 21 Modified "satellite data based" to "satellite-based"

Page 5, line 23-26 Removed all text after "(Gash, 1979)", replaced by:

(Gash, 1979) as a first step. GLEAM then employs the Priestley–Taylor equation to calculate the potential evaporation of bare soil and vegetation components (both short and tall canopy), with values constrained to actual evaporation via application of a stress factor. The stress factor is calculated using vegetation optical depth from a combination of different satellite passive microwave observations using the Land Parameter Retrieval Model (Liu et al. 2013). GLEAM also has the capacity to explicitly calculate sublimation of snow covered surfaces (Takala et al., 2011) as well as open water evaporation. Satellite observations of surface soil moisture can be assimilated using a Kalman filter assimilation approach to estimate the moisture profile over several soil layers. Here we employ version 2A of GLEAM (Miralles, 2014, personal communication), which uses a combination of satellite, ground and reanalysis input data. Precipitation is obtained from the Climate Prediction Center Unified data set, consisting of data from over 30,000 stations (CPC-Unified, Joyce et al., 2004). The radiation product used in this version of GLEAM is the European Center for Medium-Range Weather Forecasts (ECMWF) ERA-Interim meteorological reanalysis product (Dee et al., 2011). In this version of GLEAM, surface soil moisture data from the Water Cycle observation Multi-mission Strategy Climate

Change Initiative (WACMOS-CCI) merged product (from a combination of several passive and active microwave products) is assimilated (Liu et al., 2012), while air temperature is derived from both the International Satellite Cloud Climatology Project (ISCCP) and the Atmospheric Infrared Sounder (AIRS) (Rossow and Dueñas, 2011). Further details of the model can be found in Miralles et al. (2010), Miralles et al. (2011a) and Miralles et al. (2011b).

Note: Due to the addition of a new citation "Miralles et al. (2011b)" in the text above, all previous citations to "Miralles et al. (2011)" have been replaced to "Miralles et al. (2011a)"

Page 6, line 5 Inserted "and has been previously used in soil moisture- and evaporation-based analyses" before "(Crow, 2007; Miralles et al., 2011a)"

Page 6, lines 9-10 Modified text:

While runoff data was not used explicitly in the consistency analysis presented in this manuscript, simulated runoff data was compared to precipitation and evaporation observations in order to evaluate  the assumption of a relatively simple hydrological system  in the study basins.

Page 5, line 14 Inserted following text before "The version of the product…"

Although these values were not constrained with ground estimates and thus may contain biases, as noted, runoff values were only used to provide an assessment of runoff against the observed precipitation and evaporation data.

Page 6, line 16 Modified "2.5 Study basins" to "2.5 Selection of study basins"

Page 6, lines 17-18 Modified text:

The study basins were  targeted  primarily on their climate classification, with river basins in regions with a predominantly arid or semi-arid climate preferentially selected.

Page 6, line 18 Inserted text before "We employed…":

This criterion was established in order to seek a relatively simple hydrological system (i.e. constrain the range of possible hydrological interactions), thereby maximizing the conditions under which hydrological consistency between evaporation and precipitation and water storage changes might be achieved.

Page 6, line 18 Modified "We employed a Köppen classification map" to "A Köppen classification map,"

Page 6, line 19 Inserted text ", was used to identify arid and semi-arid regions." after "(Kottek et al., 2006)".

Page 6, lines 19-20 Moved "The climate criteria was to select basins with more than 50% areal extent containing any of the arid Köppen climates (BWk, BWh, BSk or BSh)." after "…(USGS)" (Page 6, line 23)

Page 6, lines 21-22 Modified "(GRDC), derived" to "(GRDC) and derived"

Page 6, line 23 Inserted text before "Secondary"

A threshold of 50% areal extent containing any of the arid Köppen climates (BWk, BWh, BSk or BSh) was used to select potential basins.

Page 6, lines 23-24 Modified text:

Secondary  criteria for  basin selection from the GRDC data set focused on size, geographical  distribution and amplitude and trends in the water storage variations.

Page 6, lines 24-25 Modified text:

In terms of the size of the basin, a smaller size would more likely satisfy the assumption of a relatively simple hydrological system. However, due to the coarse resolution of GRACE data (see section 2.1), this requirement had to be compromised. Given these considerations,  four basins were selected as focus regions of study: the Colorado  River basin (CRB) in North America, the Niger River basin (NRB) in Africa, the Aral Sea basin (ASB) in Asia and the Lake Eyre basin (LEB) in Australia (Figure 1).

Page 6, line 29 Modified "compare" to "establish"

Page 6, line 29 Modified ", i.e. a simple water budget," to "(i.e. a simple water budget)"

Page 6, line 30 Modified "pre-dominantly arid" to "predominantly arid,"

Page 7, line 2 Modified "Similarly" to "Likewise"

Page 7, line 4 Modified "as important as evaporation" to "that is  close in magnitude to evaporation"

Page 7, line 5 Added "undertaken here." after "analysis"

Page 7, line 6 Inserted text, including modified text moved from page 10, lines 6-7, 22-23, and page 11, lines 9-10, 25-26.

Even though these four basins were preselected based upon their location within dryland systems (Wang et al., 2012), they reflect a range of trends in water storage and precipitation. For example, the Colorado River basin experienced intervals of wet and dry periods, while the Niger River basin showed a small but steady increase in water storage with a clear, seasonal variability in both water storage and precipitation. Meanwhile, the Aral Sea basin experienced a significant loss of water during the study period (-8.2 mm.yr$^{-1}$), in line with the historical depletion of this inland sea in response to increased agricultural productivity (Zmijewski and Becker, 2014). The Lake Eyre basin showed a marked increase in precipitation during the end of the study period (2009-2011), with a corresponding increase in water storage during the following years, reflecting the larger scale hydrometeorological conditions affecting that region (Boening et al., 2012).

**Methodology**

Page 7, line 8 Inserted "and to ensure that a fair comparison between GRACE data and satellite products could be undertaken" after "(i.e. at sub-basin scale)"

Page 7, line 8 Modified "Futhermore, the" to "The"

Page 7, line 8 Inserted "(see Section 2.1)" after "de-striping filter"

Page 7, line 8 Modified "accounted for" to "incorporated into the analysis"

Page 7, lines 13-22 Moved and modified to Page 4, line 10 (Section 2.1 GRACE water storage anomalies)

Page 7, line 24 Inserted "up to an approximation of degree $l_{max}$" after "$C_{lm}$ and $S_{lm}$"

Page 7, line 25 Modified "In this study," to "Here"

Page 7, lines 29-30 Removed line break

Page 8, line 1 Modified "for each E data set" to "for each E product"

Page 8, line 2 Inserted "(see section 2.1)" after "GRACE TWSA data".

Page 8, line 4 Modified "The computed spherical harmonic coefficients so far are global" to "In the analysis so far, the computed spherical harmonic coefficients are global"

Page 8, line 5 Modified "needed to be" to "needs", and "regions" to "basins"

Page 8, line 8 Modified "at a basin-scale" to "at the basin scale"

Page 8, line 27 Modified "region" to "basin"

**Results**

Page 9, line 3-5 Removed "As noted earlier, an objective of this work is to determine whether the choice of different evaporation products affects the hydrological consistency analysis i.e. is the disagreement between evaporation products significant enough to impact the outcomes of the study?"

Page 9, line 5 Modified "A cursory examination" to "An examination"

Page 9, line 21 Modified "hydrological closure approach" to "hydrological consistency approach"

Page 9, line 22 Added quotes to "simple"

Page 9, line 22 Added text before "While it is …":

> Indeed, this has been demonstrated in other studies using either satellite data alone, or a combination of satellite and ground data.

Page 9, line 22 Modified "this current work" to "the current work"

Page 9, line 23 Modified "these different evaporation models" to "these different evaporation models based on hydrological closure"

Page 9, line 24 Added "(precipitation and gravity-based water storage changes) that the evaporation is" before "being compared against"

Page 9, lines 27-28 Removed "A key objective of this analysis is to assess the hydrological consistency between discrete components of the water cycle (see Figure 2)."

Page 9, line 28 Modified "To do this," to "In this section," and "examined" to "examine"

Page 9, line 29 Modified "Figures 4-7" to "Figure 4 to Figure 7"

Page 9, line 30 Removed "forming the focus of this study"

Page 10, line 1 Inserted "e.g.  do the degree correlations behave differently during wet or dry periods, or when storage changes are driven by natural or anthropogenic causes?"

Page 10, lines 1-2 Removed "For example, do the degree correlations behave differently during wet or dry periods, or when storage changes can be attributed to either natural or anthropogenic causes?"

Page 10, line 5 Modified "4.2.1 Colorado river basin (CRB)" to "4.2.1 Colorado River basin"

Note: throughout the text, we replaced all basin abbreviations (e.g. CRB as in Colorado River basin) to the full name of the basins. Only the first mention of the abbreviations were keft (i.e. in Section 2.5) as well as in the Figures.

Page 10, lines 6-7 Modified text

> The start of the study period (2003) coincided with the end of an intense multi-year drought in the Colorado River basin (Scanlon et al., 2015).

Page 10, line 10 Corrected "February 2004" to "February 2005"

Page 10, line 16 Modified "runoff in the basin: a large" to "runoff in the basin, since a large"

Page 10, line 19 Modified "Figure 4, however" to "Figure 4. However,"

Page 10, line 22 Modified "region" to "basin", "and a clear" to "with clear". Added "(see Figure 5)" after "seasonality".

Page 10, lines 23-25 Modified text:

> Over the study period, p~P~recipitation peaks between July and September, while TWSA peaks between September and November. Ahmed et al. (2014) attributed the observed increase in TWSA to an increase in precipitation in the region caused by warmer Atlantic Ocean temperatures.

Page 10, line 28 Modified "GRACE observing period, the region" to "GRACE observing period (2003-2004), the basin"

Page 11, lines 5-7 Modified text:

> Interestingly, there  seems to be less inter-degree variability compared to the other study basins. This may be related to the simpler water budget in this basin compared to that of the Colorado River and Aral Sea basins , but requires further investigation.

Page 11, line 9-11 Modified text:

> This endorheic basin continued a historical trend of water loss  during the study period, most likely caused by agricultural

activities (Zmijewski and Becker, 2014).

Page 11, line 15 Added "(see Figure 6)" after "respectively"

Page 11, line 18 Modified "region" to "basin"

Page 11, lines 21-23 Modified text:

These complications are reflected in the higher inter-degree variability (compared to the other basins). Differences in degree correlation due to the use of the three evaporation products were minimal i.e. no single evaporation product resulted in a significantly higher (or lower) hydrological consistency with precipitation and water storage anomalies.

Page 11, line 25 Modified "rainy seasons in" to "rainy seasons of"

Page 11, line 28 Added "(see Figure 7)" after "period"

Page 12, line 1 Modified "In general," to "Overall,"

[revised manuscript text omitted]

Page 13, line 23 Added new sub-heading "5.1 Challenges to implementing hydrological consistency"

Page 13, line 24 New paragraph based on modified text from Page 13, lines 10-11 and 17-18

It is, with each hydrological process presenting its own limitations, errors and retrieval challenges (McCabe et al. 2008). Therefore, irrespective of the physical constraint that the concept of hydrological consistency is based upon (i.e. mass balance), given the variability in our capacity to accurately retrieve hydrological responses via satellite-based systems, it is not unreasonable to expect that achieving hydrological consistency might be a difficult task. However, it should be reasonable to expect that in some regions, especially those where simpler and more defined water cycle behaviour dominates, that more significant and consistent inter-product agreement between hydrological components should existmight be found. For this reasonTo explore this idea, ourthe study assumed focused on basins where aa simpler water budget, consisting of water storage anomalies as a function of precipitation and evaporative fluxes, might be expected to predominate. The aim was to limit potentially complicating variables such as snow, vegetation changes, large precipitation and streamflow contributions and other hydrological processes. The assumption was that arid- and semi-arid regions would best fit this profile. only and was deliberately limited to regions that would more closely reflect this simple closure assessment as much as possible. It is worth noting that an assumption of a simplified water budget in order to evaluate agreement in satellite products has been employed before. Indeed, Wang et al. (2014) applied this concept to evaluate the level of agreement between three satellite products over arid regions in Australia, assuming surface and subsurface runoff were minimal. Given the relationship between size and accuracy for GRACE data, a geographically distributioned selection of basins that could satisfyfit into this simplified water budget assumption is somewhat limitedwas difficult. Restrictions related to basin size affect the study in two conflicting ways. On the one hand, a large basin will inevitably present complications related to heterogeneity (including in climate zones, as was the case for the Colorado River basin and Aral Ssea basin) and also be more likely to contain areas affected by anthropogenic activities, such as irrigation, land cover changes, building of dams and reservoirs, etc. On the other hand, a small catchment size would be more difficult to evaluate with this consistency approach, given the coarse resolution of (most) of the global products used here,

but especially the GRACE data. The spatial resolution of GRACE data is further limited by the use of filters to remove errors. Considering these restrictions, a compromise in the selection of study basins was required to allow for at least a narrow range of length scales (500-800 km) to be evaluated.

Page 13, lines 18-22 Modified and moved as third paragraph in the revised Discussion section:

In the end, our study consisted of four major globally distributed river basins, including two endorheic systems. Although having mostly an arid climate in terms of Köppen classification, both the Colorado River and Aral Sea basins include regions with the presence of snow and snowmelt-dominated runoff. While snow storage itself is not a problem, since GRACE detects changes in storage irrespective of their nature (snow, groundwater, soil moisture, etc), snowmelt may contribute to delayed changes in storage that can affect gravity results. As such, the inclusion of these basins was considered important in order to test the hydrological consistency concept in regions that deviated from the ideal assumption. The influence that snowmelt and other potential sources of lag in the system have is poorly defined and forms part of the reason to explore the inclusion of a lag response in the GRACE data (see Section 4.3).

Page 13, lines 24 to Page 14, line 14 Removed

Page 13, Addition of a new paragraph (fourth) based on text from Page 15, lines 12-23:

Apart from the issues of spatial scale, the use of satellite-based hydrological data presents additional challenges and sources of uncertainty to a consistency-based assessment. For instance, because GRACE data is smoothed to remove errors in short-scale terms (i.e., truncation of the spherical harmonic coefficients), the gravity signal contains contamination from outside of the studied basins (leakage) and is a potential source of uncertainty in areas neighbouring high amplitude signals (particularly if they are out of phasesingle signalwith the study basin) and the ocean. Although the LSM-based scaling factor, which is static in time, has been used to correct for bias (e.g. signal reduction) and leakage contamination, dynamic changes in water storage trends outside the basin might still contaminate the signal (Long et al., 2015). In addition, the temporal lag in terrestrial storage response as documented in previous studies (Rieser et al., 2010; Ahmed et al., 2011; Wang et al., 2014; Ayman and Jin, 2016) and observed in our analysis, wasrepresents an important source of potential error (see Sections 4.3 and 5.2). Product errors are also evident in the precipitation and evaporation data sets. Global rainfall retrievals have well recognised limitations, including the detection of both high and low intensity events (Hou et al., 2014), the discrimination of cloud clear and cloud precipitation scenes, as well as the sensitivity to parameters in the forward model of radiative transfer over different sensors (Stephens and Kummerow, 2007). In terms of evaporation, uncertainties related to algorithm choice, input data variability and process parameterizations all complicate the accurate estimation of terrestrial fluxes (Ershadi et al. 2015). Determining whether or not and understanding how much these product sources of uncertainty affect hydrological consistency studies remains an area requiring further investigation.

Page 14, lines 14- 26 Modified text; moved to new fifth paragraph under a new sub-heading "5.2 Temporal lag in terrestrial storage response":

In exploring the relationship between GRACE water storage changes and precipitation and evaporation data, it was evident that water storage anomalies peaked at a significantly later time than the corresponding P-E values. One possibility for this apparent lag in that has been explored in studies using GRACE data (Rieser et al., 2010; Ahmed et al., 2011) is that, water anomalies are

detected some time after a precipitation event, due to the inability of GRACE to detect small-scale changes in the gravity field (a rough estimate of GRACE accuracy averaged over the entire Earth is 20 mm.month$^{-1}$; Wahr and Velicogna, 2006), and therefore the corresponding mass is not detected until a sufficient amount has accumulated within the catchment via natural drainage processes (Rieser et al., 2010; Ahmed et al., 2011). The intensity and duration of the precipitation events, antecedent soil moisture, as well as the geomorphological characteristics of the basin would thus all influence the detection time. A simple way to account for this phenomenon was to apply a phase lag to GRACE data by a constant amount for the whole study period. Doing tThis seemed to improved the behaviour of degree correlation, not only in time (less variability in the results), but increased the value of $r_l$ as well. This was particularly evident in the Niger rRRiver basin, which was expected due to the well-defined seasonal behaviour of its hydrological cycle throughout the study period, and to a lesser extent in the Colorado RrRiver basin and Aral Sea basin, where changing trends in the seasonal patterns of precipitation make made it more challenging to apply this simple correction. In the Lake Eyre basin, applying a lag to GRACE data did not seem to have an effect on the degree correlation. Further understanding the implications and physical rationale behind the attribution of this lag is required.

Page 14, line 28 to Page 15, line 10 Modified text; moved to new sixth paragraph under a new sub-heading "5.3 Discriminating between satellite evaporation products"

A secondary objectiveOne aspect of this work was to determine explore whether differences in available evaporation products could affect the results of the consistency analysis i.e. could we identify was there better agreement between water storage anomalies and P-E with in anyone particular evaporation product? While tThe analysed products used in the study covered a wide range of resolutions (0.05°–0.25°), although the effective resolution in the analysis is was ultimately determined by the truncation degree ($l_{max}$) of the spherical harmonic transformation. Even after accounting for this, absolute differences were evident from a qualitative basin-scale analysis (Figure 2). Overall, rResults indicated that MOD16 underestimated evaporation when compared to CSIRO-PML and GLEAM, even though both the CSIRO-PML and MOD16 products are based on the Penman–-Monteith equation and rely heavily on MODIS data. Several recent studies (McCabe et al., 2016; Michel et al., 2016; Miralles et al. 2016) also suggest that the MOD16 product (or variants using the PM-Mu approach) underestimate evaporation when compared to other products (including GLEAM), and that most products show large discrepancies in reproducing results during periods of water stress. Ershadi et al. (2015) demonstrated that the parameterization of aerodynamic and surface resistances were critical controls on evaporation through both soil and vegetation. Furthermore, both GLEAM and CSIRO-PML include dynamic constraints on evaporation (stress module and soil moisture assimilation in GLEAM; dynamic ratio of actual to potential evaporation at the soil surface in CSIRO-PML) that are critical in arid regions due to hydrological and plant physiological stresses and the subsequent importance of soil evaporation. Whether these differences in model parameterization are the sole cause of the apparent underestimation by MOD16 remains to be investigated.

Page 15, lines 12-23 Moved and extensively modified to serve as the fourth paragraph in the revised Discussion Section (see text above)

Page 15, line 25 to Page 16, line 7 Extensively modified and separated into two new paragraphs:

[revised manuscript text omitted]

**Conclusion**

Page 16, lines 9-10 Modified "one of the motivating elements of this study" to "a key one of the motivation behindng elements of this study"

Page 16, line 10 Removed "(or estimates)"

Page 16, line 11 Modified "to achieve" to "to reflect", and "closure" to "consistency"

Page 16, line 11 Modified "To do this" to "To do so"

Page 16, line 12 Modified "hydrological consistency" to "this"

Page 16, line 12 Modified "that is, in arid regions" to "in arid and semi-arid regions"

Page 16, line 12 Modified "water budget consisting (ideally) of only" to "water budget, consisting primarily of"

Page 16, line 13 Modified "We found" to "Unfortunately, we found"

Page 16, line 14 Removed "throughout the study period"

Page 16, line 16 Modified "in individual products" to "within individual products"

Page 16, line 16 Removed "Furthermore,"

Page 16, line 17 Modified "significant differences" to "significant and known differences"

Page 16, line 18 Modified "did not play" to "did not seem to play"

Page 16, lines 18-20 Moved and modified text to next paragraph:

> While  not providing a comprehensive tool for product evaluation, the approach did help to reveal some interesting spatial and temporal patterns between the studied hydrological variables.

Page 16, line 25 Modified "then" to "with"

Page 16, line 27 Added "quite" before "reasonable"

Page 16, line 28 Added "to account for delayed sources of water storage changes" after "considered"

Page 17, line 1 Modified "in other regions" to "in some of the studied basins"

Page 17, line 7 Modified "as a potential tool in" to "as an element of a holistic approach to "

Page 17, line 26 Added new references:

[revised manuscript text omitted]

---

## Referee Report (RR1)

SUMMARY

This revised manuscript reads infinitely better that the first version. My compliments for that!

This paper addresses a valid tool, not new but new to this application, to analyse the consistency of water variables. Given all the difficulties to address this inconsistency, encountered in other studies, it is therefore worth publishing. However, the conclusion that 'large-scale evaporation products are inconsistent', both among themselves as with other water budget components, needs more clarification. After reading through it a couple of times, I think the main message is to make sure that there is a discussion or conclusion that the problem of inconsistency does not necessarily lie in the satellite products, but in their underlying input components. Therefore, I advise 'acceptation with major revision'.

MAJOR COMMENTS

Even though the authors test a new comparison tool, one has to be aware of what is compared. In my opinion, this analyses could have gone a bit deeper and explain what the origin of the inconsistencies could be (for example, input meteo data). Let me explain with two examples:

1) Differences caused by the different ET data can either be caused by the method, or their input data. In the case of MOD16, for example, input data comes from the GMAO data. But correlations are made with the GPCP data. How do the GPCP data and the GMAO data correlate, for example in terms of rainfall, RH etc. That can be a cause for bad correlation. In my opinion, such causes need to be mentioned, because it could well be that the satellite observations of one method are better than another, but the other input data messes up the correlation. In my opinion, this needs to be discussed, even with an example.
2) Following from 1), in my opinion, one should have better compared the GPCP data with some components of the GMAO dataset. If inconsistencies can be found there, they can explain inconsistencies between MOD16 and other products not based on GMAO.

Therefore, this paper reads a bit like 'we have a really cool tool (which it really is!) and we use it compare water variables'. But for me, the conclusion that there is inconsistency that needs to be worked on (although a true statement) sounds a bit too easy. What are these underlying causes? Are they actually caused by the satellite data? Or by the model input components? This needs to be properly discussed.

Another point, but similar, that still worries me about Figs 4-7 is the outflow part. Again, this question pops to my mind: "are we comparing the right products?". Any remaining P-ET will either come out of the catchment as quickflow or baseflow. Baseflow can take months to years to come to the surface, whereas quickflow can leave the catchment in days. Is GRACE data corrected for outflow and how? If it is, ok, stop reading, my bad. If not, GRACE does not only show the inflow (P-ET), but also the outflow. And this TWSA needs to compared to P-ET-Q (i.e., Q being all outflow out of the catchment). This means that Figs 4-7 could either show an inconsistency in measured water volumes, or the difference between outflow and inflow, or (probably) a combination of both.

I am not saying that the above points need to be part of this paper (although it would be nice), but they need to be at least properly addressed in the discussion, conclusion and abstract.

MINOR COMMENTS

Only two typos found:

P1: line 15: after 'environments', remove 'including' and put the basin names in between brackets.

P2, line 3: 'wide range', not 'wide-range'.

---

## Author Response (AR2)

Dear Prof. Dr. Marc Bierkens,

We thank you for your efforts in assessing our manuscript. We have examined the reviewer's questions regarding (1) the causes for inconsistencies and (2) the interpretation of the outflow component in the analysis. As a result, we have adapted some key points throughout the manuscript and added some additional clarification and explanations, particularly in the Discussion and Conclusion sections.

To address the first point, we have restated in the revised Introduction that we are inter-comparing "official" evaporation products rather than evaporation models. This distinction is important, because the aim of the study is to evaluate how well do these "end-user" products integrate with other related hydrological variables. The key (and somewhat surprising) finding of the study was indeed that there is considerable inconsistency. Establishing the causes, either as a consequence of the individual product uncertainties (including also the precipitation and GRACE products), from the input data forcing the evaporation model, or some other factor, will require ongoing investigation.

As previous studies (e.g. Vinukollu et al., 2011; Ershadi et al., 2014; McCabe et al., 2016; Miralles et al., 2016) have already inter-compared many evaporation models, as well as the role of input forcing variability on these models, demonstrating their intra-product variability, we did not undertake further such analysis. However, in our response to the reviewer, we have included an analysis on the correlation between the precipitation product used in the hydrological consistency assessment (GPCP) and the precipitation data that was used in two of the evaporation products (GPCC and CPC-Unified). We determined that the products were sufficiently similar in terms of the same measure used in the study (degree-correlation), as well as with the pattern correlation, a measure of spatial correlation (Walsh and McGregor, 1997). It was concluded therefore that the difference in input precipitation does not fully explain the inconsistencies found in the study. It is important to note that the absolute differences found in the evaporation products used in our study did not represent a major impact on the comparison of the products in terms of hydrological consistency. Therefore, the issue of consistency not being observed is still somewhat undetermined, and will require further examination on product quality and understanding what some of these hydrological retrievals are actually expressing.

As discussed in the response to the reviewer's comments, efforts to minimize the impact of the outflow component (such as selecting smaller catchments) were hindered by the spatial limitations of using GRACE data. As such, it was not possible to fully exclude this component. However, the measure used in the main analysis of our study (i.e. the degree-correlations) evaluate a measure of spatial agreement between the inflow and outflow, assuming that the errors in the outflow component do not affect these spatial patterns (or that they are dominated by changes in storage, precipitation and evaporation). We have added some clarification in the body of the manuscript to further describe our assumptions regarding the outflow component in Figures 4-7 e.g., *"In these figures, the degree correlation is a measure of the spatial agreement between the two fields being compared (i.e. TWSA and P-E anomalies), assuming that other outflow components such as surface runoff (assumed to be minimal in these basins) do not directly affect these spatial patterns."*

Overall, we believe that this work represents a novel and interesting study on the apparent inconsistency between commonly used hydrological products. Clearly, further work will be required to disentangle the underlying reasons for this, but it represents an area of some concern in establishing product quality. Additionally, drawing attention to the need to more thoroughly evaluate remote sensing based products using a range of metrics is also a worthwhile outcome, as the current paradigm of point to pixel evaluation is often inadequate. This is especially true for the global, long-term datasets that are increasingly becoming available to the community.

We thank you again for your time and consideration.

Best wishes,

Oliver Lopez, Rasmus Houborg and Matthew McCabe

**Author's response to Rogier Westerhof**

We thank Rogier Westerhof for his time and attention in reviewing this paper on both occasions. We have made an effort to address his new comments, which are included below, along with our responses (*in italic*).

This revised manuscript reads infinitely better that the first version. My compliments for that!

This paper addresses a valid tool, not new but new to this application, to analyse the consistency of water variables. Given all the difficulties to address this inconsistency, encountered in other studies, it is therefore worth publishing. However, the conclusion that 'large-scale evaporation products are inconsistent', both among themselves as with other water budget components, needs more clarification. After reading through it a couple of times, I think the main message is to make sure that there is a discussion or conclusion that the problem of inconsistency does not necessarily lie in the satellite products, but in their underlying input components. Therefore, I advise 'acceptation with major revision'.

*Thank you for your thoughtful comments. To clarify, an actual implementation of the hydrological consistency concept remains quite new and novel, especially in exploring ways to evaluate agreement in satellite products. To our knowledge, this is the first application using the degree correlation approach in spherical harmonics for this purpose and the first to assess gravity and rainfall against a number of "competing" evaporation products. We believe it presents an interesting and unique perspective on the product evaluation challenge and appreciate the reviewer's positive perspective on this problem.*

*Within our study, we did not aim to inter-compare the underlying forcing used in the different methods, but rather make use of "official" products, as this is what a user is most likely to adopt i.e. an off-the-shelf or available product, rather than developing one of their own. The use of the terms "product" and "model" has been revised in the text to better reflect these differences (see for example, the use of the word product in page 2, lines 26 and 28; now lines 28 and 30, referring to a number of previous studies). This has also been indicated in the Abstract (page 1, line 16): ",* based on different methodologies and input data".

*The conclusion that these 'large-scale evaporation products are inconsistent' remains the key finding: they are not only inconsistent amongst themselves (as our earlier intercomparison papers have found – see comment below), they are inconsistent with related hydrological variables. What is the cause of this inconsistency? We suspect it is as much the individual products (P, E and GRACE data) as it is the "input" components forcing the evaporation retrievals. It is also worth keeping in mind that the product variability (in terms of evaporation) is not simply a function of different input forcing: there is the choice of algorithm, implicit model assumptions, choice of parameterizations and performance variability as a function of land cover, that all impact retrieval accuracy.*

*We agree that all of these elements play a part in the issue of inconsistency and have adapted some text in the Discussion section (see below) to reflect this: but it is still troubling that amongst the products being examined, there are none that reflect what could be termed a strong reproduction of consistency, even considering the observed intra-product differences.*

*Change to Discussion section in Page 17, line 9 (Page 18, line 1 in revised manuscript): "However, other sources of errors that directly affect evaporation estimates, such as the choice of algorithm, implicit model assumptions, choice of parameterizations and an incorrect representation of the land cover, can directly impact the degree correlation measure."*

Even though the authors test a new comparison tool, one has to be aware of what is compared. In my opinion, this analyses could have gone a bit deeper and explain what the origin of the inconsistencies could be (for example, input meteo data). Let me explain with two examples:

1) Differences caused by the different ET data can either be caused by the method, or their input data. In the case of MOD16, for example, input data comes from the GMAO data. But correlations are made with the GPCP data. How do the GPCP data and the GMAO data correlate, for example in terms of rainfall, RH etc. That can be a cause for bad correlation. In my opinion, such causes need to be mentioned, because it could well be that the satellite observations of one method are better than another, but the other input data messes up the correlation. In my opinion, this needs to be discussed, even with an example.

2) Following from 1), in my opinion, one should have better compared the GPCP data with some components of the GMAO dataset. If inconsistencies can be found there, they can explain inconsistencies between MOD16 and other products not based on GMAO.

Therefore, this paper reads a bit like 'we have a really cool tool (which it really is!) and we use it compare water variables'. But for me, the conclusion that there is inconsistency that needs to be worked on (although a true statement) sounds a bit too easy. What are these underlying causes? Are they actually caused by the satellite data? Or by the model input components? This needs to be properly discussed.

*Some of these issues are raised in the response to the reviewers first comment, so we do not repeat them here (i.e. relating to input forcing being well studied in related papers, but only being one element of uncertainty in the retrieval products).*

*In recent years, there have been a number of efforts to compare the underlying methods and algorithms of satellite evaporation retrieval, including the impact of varying sources of input data and at different resolutions that have been led both by our own group and other collaborators (see Vinukollu et al., 2011; Ershadi et al., 2014; McCabe et al., 2016; Miralles et al., 2016). Given these previous studies, we did not undertake a further inter-comparison in this study, since the intra-product variability is well recognised. Although there were indeed some absolute differences in the evaporation products (as shown in Figure 2) that are caused by either differences in the models or input data, these did not ultimately represent an advantage or disadvantage between the products in terms of their hydrological consistency.*

*We were hoping that the consistency approach might offer a way to identify which of these products might be "better" than the others. What we found was that there was no strong consistency case for any of the products – regardless of, rather than in spite of, their underlying differences. Restating our response to the first comment: it is troubling that consistency has not been observed. Getting to the bottom of this will require considerably more investigation, both in terms of improving product quality, but also in understanding what some of these hydrological retrievals are actually expressing. It may also be a case that the approach is invalid: but this raises some questions of its own, given the relatively fundamental basis upon which it is derived.*

*The reviewers question on why the GPCP data was used in the study is important. The GPCP product was selected based on the same requirements used to select the evaporation products: global availability and mostly satellite-based. Indeed, two of the evaporation products use precipitation as input: GLEAM has an interception module that requires precipitation as input (in the version we have used, it uses the CPC-Unified precipitation product: Joyce et al, 2004), while CSIRO-PML uses precipitation data (GPCC: Becker et al., 2013) to run a hydrometeorological model in the calibration of its spatial parameters. MOD16 does not directly use precipitation and we have added this clarification to the text. To address the concern though, we have now examined the agreement between these three precipitation products, both in*

*the spatial domain as pattern correlations (Walsh and McGregor, 1997) and using the same measure employed in the study (degree correlation in spherical harmonics; Arkani-Hamed, 1998). The agreement between these precipitation products is consistently high (see Figure R1 below), while the agreement between GRACE and P-E shown in our study was not. Therefore, the difference in input precipitation products does not explain the inconsistencies found in the study.*

Another point, but similar, that still worries me about Figs 4-7 is the outflow part. Again, this question pops to my mind: "are we comparing the right products?". Any remaining P-ET will either come out of the catchment as quickflow or baseflow. Baseflow can take months to years to come to the surface, whereas quickflow can leave the catchment in days. Is GRACE data corrected for outflow and how? If it is, ok, stop reading, my bad. If not, GRACE does not only show the inflow (P-ET), but also the outflow. And this TWSA needs to compared to P-ET-Q (i.e., Q being all outflow out of the catchment). This means that Figs 4-7 could either show an inconsistency in measured water volumes, or the difference between outflow and inflow, or (probably) a combination of both.

*The basins were pre-selected to try and minimize the outflow component – which the reviewer correctly identifies as an issue. Due to the catchment size restrictions that come with using GRACE data, it is not possible to entirely discount this component. Even though the selected basins are located in predominantly arid regions (where runoff is not expected to be a significant variable), they do include regions with outflow components in the form of snowmelt, surface and subsurface runoff. This will certainly impact the water budget closure in terms of measured water volumes. However, we are not evaluating the total basin water closure. Instead, we are evaluating a measure of spatial agreement between the inflow and outflow, assuming that the errors in the outflow component will not dramatically affect these spatial patterns. We have added the following text in the Discussion section (Page 17, line 9; now Page 18) to clarify this: "The role of the degree correlation was to evaluate the spatial agreement between the hydrological components, assuming that any non-closure errors due to unmodeled outflow components (e.g. long-term baseflow or minimal surface runoff) would not affect this measure. However, other sources of errors that directly affect evaporation estimates, such as the choice of algorithm, implicit model assumptions, choice of parameterizations and an incorrect representation of the land cover, can directly impact the degree correlation measure."*

*We have also added the following text in the "Basin scale assessment" sub-section (4.2; page 12, line 23): "In these figures, the degree correlation is a measure of the spatial agreement between the two fields being compared (i.e. TWSA and P-E anomalies), assuming that other outflow components such as surface runoff (assumed to be minimal in these basins) do not directly affect these spatial patterns."*

*In the conclusions section (page 20, line 9), the following text was modified as well to clarify the missing outflow component: "To do this, the study focused on regions where it would be most expected to observe such responses: arid and semi-arid regions with a simplified water budget, consisting primarily of precipitation and evaporation, and assuming a minimal runoff and other long-term outflow components."*

I am not saying that the above points need to be part of this paper (although it would be nice), but they need to be at least properly addressed in the discussion, conclusion and abstract.

*We do not disagree with the reviewer on many of the points being raised. Where possible, we have tried to better address or clarify these areas of concern throughout the paper, as they reflect key implications of the work. Certainly there is scope to expand this work to further disentangle some of these underlying issues. Referring to this last point, the following text was added to the Conclusion section to better describe this need: "Despite these challenges, the expectation is that retrievals of global and regional products will inevitably improve with advances in resolution, process understanding and forcing data accuracy. In*

*concert with such product improvements, the way in which we evaluate remotely sensed variables should also evolve beyond the relatively simplistic comparisons against in-situ data that form the basis of most current assessments. Such a strategy would include evaluation against related hydrological variables, reflecting the underlying rationale of hydrological consistency and hydrological closure studies. Only by implementing a more comprehensive evaluation framework into our assessment schemes will greater confidence in component retrievals be realized."*

[Figure]

Figure R1. Average degree correlation (SH), average pattern correlation (ALL) and seasonal pattern correlations (DJF, MAM, JJA and SON) between the precipitation products GPCP, GPCC and CPC over the studied basins: Colorado River basin (CRB), Lake Eyre basin (LEB), Aral Sea basin (ASB) and Niger River basin (NRB). The basins were ordered by decreasing number of gauge stations.

**List of relevant changes to the manuscript**

Page 1, line 16: Added "based on different methodologies and input data," before "including CSIRO-PML"

Page 2, line 25: Modified text

Some recent evaluation efforts have sought to estimate the uncertainty of satellite-based  evaporation products, as well as those from land surface model and reanalysis data, in terms of the variance amongst the products (Mueller et al., 2011; Jimenez et al., 2011; Long et al., 2014). These and related attempts have shown that no single evaporation model consistently outperforms any other, whether applied at local (Ershadi et al., 2014) or global scales (Miralles et al., 2016.), even in cases where the same input data has been used.

Page 6, line 27: Added ",precipitation" after "wind speed"

Page 12, line 19: Added "In these figures, the degree correlation is a measure of the spatial agreement between the two fields being compared (i.e. TWSA and P-E anomalies), assuming that other outflow components such as surface runoff (assumed to be minimal in these basins) do not directly affect these spatial patterns." after "degree correlation ($r_l$)."

Page 16, line 30: Added "The role of the degree correlation was to evaluate the spatial agreement between the hydrological components, assuming that any non-closure errors due to unmodeled outflow components (e.g. long-term baseflow or minimal surface runoff) would not affect this measure. However, other sources of errors that directly affect evaporation estimates, such as the choice of algorithm, implicit model assumptions, choice of parameterizations and an incorrect representation of the land cover, can directly impact the degree correlation measure." after "would best fit this profile".

Page 17, line 14: Added "Likewise, evaporation models generally have a difficult time adequately estimating sublimation." after "gravity results"

Page 17, lines 32-34: Modified to "McCabe et al. 2016 present a thorough description of accuracy issues related to global products. However, it is not the intent of this work to explore these product uncertainty issues in detail. Determining whether or not and understanding how much these  sources of product uncertainty affect hydrological consistency studies is an important area requiring further investigation. What is clear from this analysis is that there is still some way to go in terms of being able to confidently assert that any single global product outperforms any other: at least in terms of its inter-product consistency."

Page 18, line 32: Added "However, these differences in absolute values did not affect the overall results of degree correlation dramatically, i.e. we could not identify a significant advantage or disadvantage in terms of hydrological consistency. " after "remains to be investigated".

Page 19, line 31: Added: ", and assuming a minimal runoff and other long-term outflow components." after "precipitation and evaporation."

Page 20, line 4: Added: ", whether caused by model parameterizations or by input data," after "these differences".

Page 20, line 17: Added "Implementing techniques to better account for these delayed sources of outflow could prove highly beneficial for the analysis of hydrological consistency." after "results in time".

[revised manuscript text omitted]